# The master male sex determinant Gdf6Y of the turquoise killifish arose through allelic neofunctionalization

Annekatrin Richter [1] ✉, Hanna Mörl [1], Maria Thielemann[1,4],
Markus Kleemann[1,5], Raphael Geißen [1,6], Robert Schwarz [1], Carolin Albertz[1],
Philipp Koch [1], Andreas Petzold[1,7], Torsten Kroll [1], Marco Groth [1],
Nils Hartmann[1,8], Amaury Herpin [2] & Christoph Englert [1,3] ✉

Although sex determination is a fundamental process in vertebrate development, it is very plastic. Diverse genes became major sex determinants in teleost fishes. Deciphering how individual sex-determining genes orchestrate sex determination can reveal new actors in sexual development. Here, we demonstrate that the Y-chromosomal copy of the TGF-β family member *gdf6* (*gdf6Y*) in *Nothobranchius furzeri*, an emerging model organism in aging research, gained the function of the male sex determinant through allelic diversification while retaining the skeletal developmental function shared with the X-chromosomal *gdf6* allele (*gdf6X*). Concerning sex determination, *gdf6Y* is expressed by somatic supporting cells of the developing testes. There it induces the male sex in a germ cell-independent manner in contrast to sex determination in zebrafish and the medaka. Looking for downstream effectors of Gdf6Y, we identified besides TGF-β signaling modulators, especially the inhibitor of DNA binding genes *id1/2/3*, the mRNA decay activator *zfp36l2* as a new GDF6 signaling target.

The African turquoise killifish *Nothobranchius furzeri* with its short lifespan and yet remarkable aging phenotypes is an attractive model organism, especially in aging research[1]. The species inhabits ephemeral ponds from which several laboratory strains with varying lifespans were derived e.g., GRZ from Southern Zimbabwe[2] or MZM0403 and MZM0410 from Mozambique[3,4] with lifespans of a half or one year, respectively. The embryos survive the dry season in the wild by facultative developmental arrests named diapauses. In direct development without diapause, the embryos enter the pharyngula stage and continue organogenesis after 7 days post-fertilization (dpf), which is completed at around 14 dpf[5]. The embryos are ready to hatch around

three weeks after fertilization and this so-called hatching stage (0 days post-hatching – 0 dph) can be maintained for several weeks. After hatching the larvae quickly grow, reach sexual maturation at three to four weeks post-hatching, and develop a strong sexual dimorphism including metabolic differences, particularly in the liver as shown in adult animals six weeks post-hatching[6]. With a 1:1 sex ratio, males were found to be the heterogametic sex, and an XY sex-determination system was suggested[7]. When the species' genome was sequenced, the sex chromosome pair was identified[8,9] and the Y-chromosomal allele of the growth differentiation factor 6 gene (*gdf6*), which was therefore named *gdf6Y*, was proposed as the potential sex-determining (SD) gene[8].

[1]Leibniz Institute on Aging - Fritz Lipmann Institute (FLI), Jena, Germany. [2]INRAE, UR1037 Laboratory of Fish Physiology and Genomics, Campus de Beaulieu, Rennes, France. [3]Institute of Biochemistry and Biophysics, Friedrich Schiller University Jena, Jena, Germany. [4]Present address: BianoGMP GmbH, Gera, Germany. [5]Present address: Abbott Rapid Diagnostics Jena GmbH, Jena, Germany. [6]Present address: Memorial Sloan Kettering Cancer Center, New York, NY, USA. [7]Present address: DRESDEN-concept e. V., Technical University (TU) Dresden, Dresden, Germany. [8]Present address: Institute of Pathology, University Medical Center of the Johannes Gutenberg University Mainz, Mainz, Germany. ✉e-mail: Annekatrin.Richter@leibniz-fli.de; Christoph.Englert@leibniz-fli.de

Gdf6 is a secreted ligand-protein that belongs to the bone morphogenetic proteins (BMPs), a group of the transforming growth factor beta (TGF-β) superfamily. These proteins consist of an N-terminal prodomain, which is proteolytically cleaved off upon maturation, and the C-terminal ligand portion, which binds as a dimer to members of the corresponding TGF-β receptor family. This leads to the intracellular activation of receptor-regulated SMAD transcription factors (R-SMADs) by phosphorylation and ultimately regulates gene expression. Other TGF-β ligand and receptor family members, including orthologs of *amh*, *amhrII*, *bmprI*, and *gsdf*, are involved in sexual development in vertebrates and were previously identified as SD genes in various fish species[10]. Therefore, factors involved in TGF-β signaling constitute one functional group, out of which SD genes are recruited in fishes. The other main group of SD gene candidates includes transcription factors expressed by the predecessors of the gonadal somatic supporting cells, which are Sertoli cells in males and granulosa cells in females. So far, this group comprises *dmrt1* and *sox3*, which were known to be involved in vertebrate sex determination before being discovered as SD genes in fish, as well as *irf9*, which is the ancestor of the rainbow trout's (*Oncorhynchus mykiss*) SD gene *sdY* but surprisingly had no previously described gonadal function[11].

In the sequenced *N. furzeri* strain GRZ, *gdf6Y* differs from the X-chromosomal *gdf6* copy, hereafter called *gdf6X*, in 22 single nucleotide variants (SNVs) and a 9-bp deletion in the coding sequence[8] (CDS; Fig. 1a; Supplementary Fig. 1a). The non-synonymous SNVs and the 9-bp deletion are conserved among the *N. furzeri* strains analyzed so far. The size of the genomic SD region (SDR), which accumulated SNVs between the X and Y chromosomes, however, varies greatly from 196 kb to 37 Mb between the strains MZM0403 and MZM0410, respectively, indicating evolutionarily young sex chromosomes[8]. In zebrafish, *Xenopus laevis*, mice, and humans, *GDF6* orthologs were shown to be involved in joint development and eye formation[12–16]. Furthermore, its involvement in vascularization[17] and melanocyte development was observed[18]. In contrast to most of the previously identified SD genes, no sexual developmental functions have been shown for *GDF6* orthologs so far, making it an unexpected SD gene candidate[10,11].

We demonstrate that *gdf6Y* is the SD gene in *N. furzeri* by inactivating this Y-chromosomal allele leading to a complete male-to-female sex reversal. Conversely, the homozygous inactivation of the X-chromosomal allele *gdf6X* led to a skeletal phenotype. However, *gdf6Y* compensates for the loss of *gdf6X* and, therefore, must have additionally acquired the SD gene function. In this role, *gdf6Y* is expressed in the male gonad by a subset of somatic cells at 0 dph and Sertoli cells in adulthood. At 0 dph, however, oogenesis supported by the oogenesis-initiating germ-cell (GC) factor *foxl2l*[19–21] has already begun in the female gonads, and, therefore, the sex in *N. furzeri* must be determined beforehand. The analyses of *gdf6Y* transgenic, mosaic animals revealed that the gene's SD function is GC-independent and, hence, should rely on the Gdf6Y-signaling response in the testicular somatic supporting cells. The genes directly regulated in response to Gdf6Y were identified in a cell culture-based approach with *id1* and *zfp36l2* as promising candidates for involvement in sex determination. In summary, we provide experimental evidence including loss-of-function and gain-of-function approaches showing that the proposed sex-determining gene *gdf6Y* acts as the master determinator of the male sex in the turquoise killifish.

## Results

### Inactivation of *gdf6Y* leads to a full male-to-female sex reversal

Being the only annotated gene at the peak of Y-specific sequence variations in the *N. furzeri* strain with the smallest SDR (MZM0403) and the gene with the strongest signal of positive selection in the SDR of the hereafter used, genome-sequenced strain GRZ, *gdf6Y* was proposed as the SD gene in *N. furzeri*[8]. To test whether *gdf6Y* is the male

sex determinant in *N. furzeri*, we employed Cas9 and two sgRNAs specifically targeting three non-synonymous or one synonymous SNV in the 2nd exon of *gdf6Y* (Fig. 1a). This approach led to the highly improbable result of 16 female and no male F0 fish (probability: $1.53 \times 10^{-5}$; Fig. 1b). Eight of these 16 females had XY chromosomes with an average mutation rate of 97% in *gdf6Y*, as determined by cloning the heterogeneous PCR products derived from the target locus and sequencing several clones (Supplementary Fig. 1b–d). In contrast, the eight animals with XX chromosomes had no mutations in *gdf6X* that were detectable with T7 endonuclease I or Sanger sequencing (Supplementary Fig. 1b, e, f), indicating that the inactivation of *gdf6Y* was highly efficient and specific.

The thus generated, phenotypically female animals with XY chromosomes and an impaired *gdf6Y* are hereafter called phenofemales (XY*). Dissection of a mosaic F0 phenofemale revealed gonads histologically indistinguishable from XX ovaries (Fig. 1c) and 7 out of 8 F0 phenofemales produced eggs when co-housed with males. The F1 offspring raised from 5 of these F0 breeding pairs consisted of males (XY), females (XX), and phenofemales (XY*) in roughly equal parts at >28 dph. The expected quarter of animals with two Y chromosomes (YY*) was missing among the respective offspring (Fig. 1d), which is improbable assuming Mendelian ratios (probability: $2.39 \times 10^{-8}$). Among the F1 generation, different frameshifting indels leading to early stop codons in *gdf6Y* or in-frame deletions were identified to cause the phenofemale phenotype with a 9 bp deletion in the CDS of the prodomain being the most subtle mutation observed (Fig. 1e; Supplementary Fig. 2a–d; Supplementary Table 1). Furthermore, F1 individuals and animals of the two propagated phenofemale lines (Fig. 1e) were used to analyze gonadal gene expression revealing that phenofemales had significantly reduced *gdf6Y* transcript levels compared to males and resemble females in the expression of sex-specific marker genes (Fig. 1f, g; Supplementary Fig. 2e, f). Phenofemales of the propagated lines also showed no difference in fertility compared to their female XX siblings regarding the number and viability of weekly collected eggs and their survival one week after collection (Supplementary Fig. 2g, h). Taken together, the inactivation of *gdf6Y* led to a full male-to-female sex reversal.

### YY* embryos develop anomalies and die until the hatching stage

Given the discrepancy between the lack of YY* animals among the sexually mature offspring of phenofemales (Fig. 1d) and the near complete egg survival 1 week after collection (Supplementary Fig. 2g, h), we monitored the development of the phenofemales' clutches further and observed malformations of the eye and tail within the egg during organogenesis in about one-quarter of the embryos (Fig. 1h, i). Those embryos died before hatching and were identified as the YY* animals (Supplementary Fig. 3a). The YY* typical malformations were characterized by lacking tail fin development and pigmentation as well as incomplete closure of the optic fissure leading to a coloboma-like phenotype (Fig. 1i; Supplementary Fig. 3b). A histological comparison with normally developing animals at 0 dph also revealed a decreased thickness of retinal layers in the YY* embryos' eyes (Supplementary Fig. 3c).

The detrimental phenotype of the YY* embryos could be caused by mutations in or a transcriptional dysregulation of genes on the Y compared to the X chromosome. As the phenofemales were generated in the genome-sequenced strain GRZ, we investigated whether the 339 genes annotated in the strain's 26.1 Mb large SDR (sgr05: 15,031,832–41,162,746)[8] have a sex chromosome copy number-dependent regulation by consulting previously published transcriptome data from XX and XY embryos at 10 dpf (Supplementary Data 1)[8]. The expression of the most significantly up- and down-regulated genes in XX compared to XY embryos within the SDR was analyzed in a phenofemale's offspring at 11 dpf including XX, XY, XY*, and YY* embryos to examine if the genes' expressions correlate with

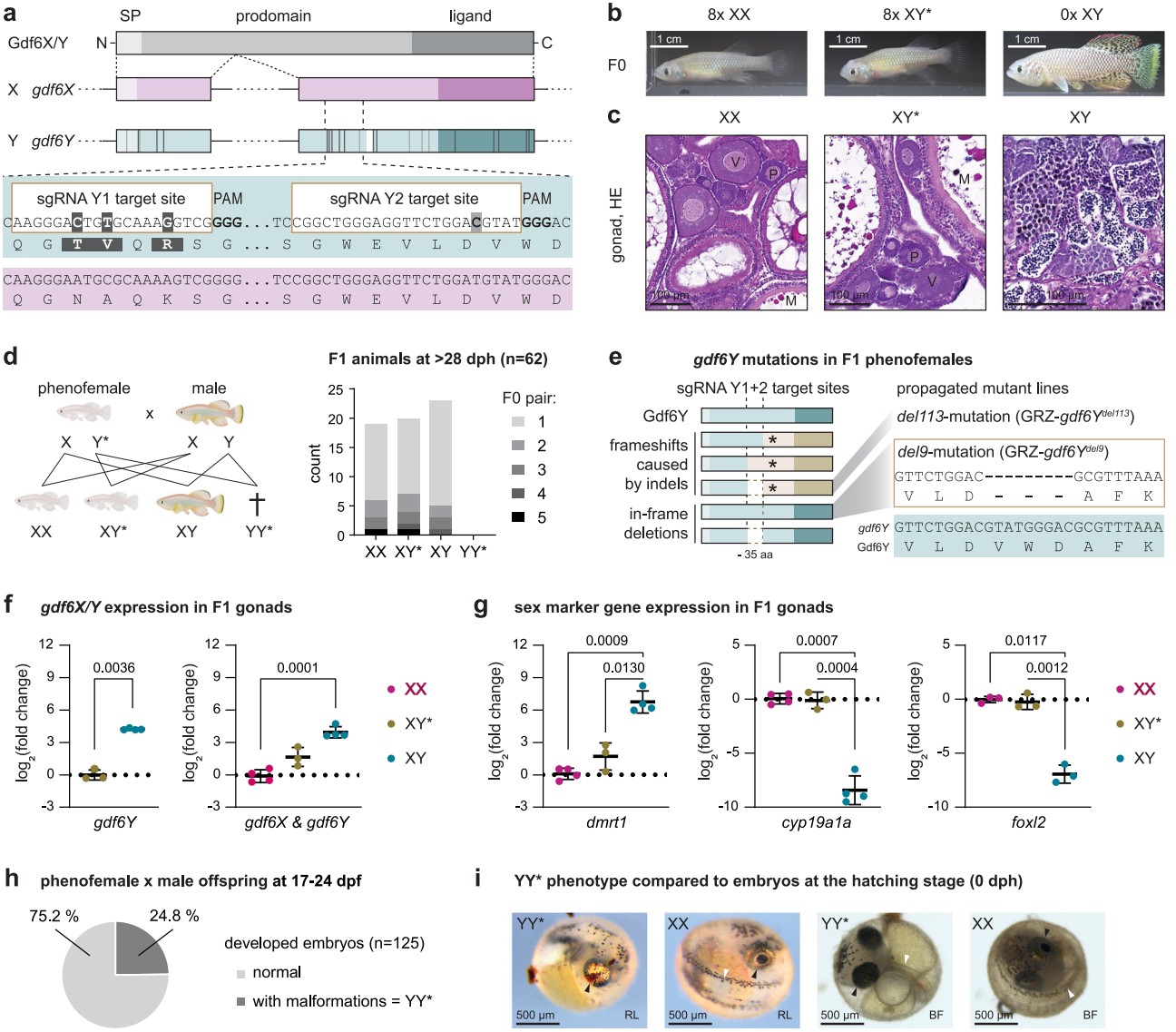

**Fig. 1 | The inactivation of *gdf6Y* in *Nothobranchius furzeri* leads to a full male-to-female sex reversal. a** Schematic of Gdf6X/Y protein domains (SP – signal peptide), exonal distribution, the coding sequence comparison between *gdf6X* and *gdf6Y*, and the mutation strategy with two *gdf6Y*-specific sgRNAs facilitating DNA double-strand breaks at a 109 bp distance. Coding sequence differences between *gdf6X* and *gdf6Y* are indicated in *gdf6Y*'s exons. SNVs: Dark gray – non-synonymous, light gray – synonymous. White – 9 bp deletion. **b** Phenotype of mosaic F0 GRZ-*gdf6Y⁻* animals (XY*, phenofemales) compared to unaltered F0 females (XX) and wild-type males (XY) at about 3 months of age. **c** One exemplarily HE stained F0 XX, F0 XY*, and XY gonad at about 5 months of age. Oocytes: P – previtellogenic, V – vitellogenic, M – mature. SC – spermatocytes, ST – spermatids, SZ – spermatozoa. **d** Sexually mature offspring (age: 1 month) of phenofemales with males consists of females, phenofemales, and males in equal parts missing YY* animals expected due to Mendelian ratios. Schematic created in BioRender. Richter, A. (2024) https://BioRender.com/n01l529. **e** Gdf6Y mutant variants causing full male-to-female sex reversal in F1 (frameshifts at either sgRNA position causing early stop codons (asterisks), in-frame deletions of 9 bp at the sgRNA Y2 site or between both sgRNAs). Two mutations were propagated in fish lines. RT-qPCR of (**f**) *gdf6Y* separately and together with *gdf6X* and (**g**) male and female marker genes in F1 gonads (Supplementary Table 1, XY*: *n* = 3; XX, XY: *n* = 4, *n* = 3 for *foxl2*) at 2.4 months of age (mean with standard deviation). Statistical testing by a two-tailed Welch's *t*-test (*gdf6Y*) or Welch's ANOVA and Dunnett's T3 multiple comparisons test (others). *P*-values < 0.05 are displayed. **h** One-quarter of a phenofemale's (GRZ-*gdf6Y^del9^*) offspring with males develops malformations and carries two Y chromosomes (YY*). **i** Reflected light (RL) and brightfield (BF) microscopy images of YY* embryos (RL, *n* = 13; BF, *n* = 31) within the egg compared to normally developed animals (RL, *n* = 45; BF, *n* = 94; here: XX at 0 dph) from independent samplings from two phenofemales (RL, GRZ-*gdf6Y^del6, del8^*; BF, GRZ-*gdf6Y^del9^* from **h**). Arrows: Black – eye, white – tail. **d**, **f**–**h** Source data are provided as a Source Data file.

the number of X or Y chromosomes being two, one, or none in the respective samples (Supplementary Fig. 3d). The most significantly upregulated gene in the XX group, *rnf19a*, showed an X chromosome-dependent expression. While the correct PCR product was amplified during RT-qPCR in YY* embryos, indicating *rnf19a* had not been lost on the Y chromosome, its massive downregulation in this sample group suggests Y-chromosomal silencing. The most downregulated gene in the XX group, *pctp*, showed an increased expression only in the presence of one or two Y chromosomes suggesting the acquisition of an

enhancer in its proximity. These two examples show that transcriptional dysregulation due to Y inactivation by silencing or sequence changes on the Y compared to the X chromosome could cause or contribute to the YY* phenotype.

## *Gdf6Y* compensates for the loss of *gdf6X*

As *GDF6* orthologs are known to be involved in eye, joint, and melanocyte formation in various species[12,13,16,18], we wondered whether the lack of *gdf6X* in YY* embryos contributes to their malformations. To

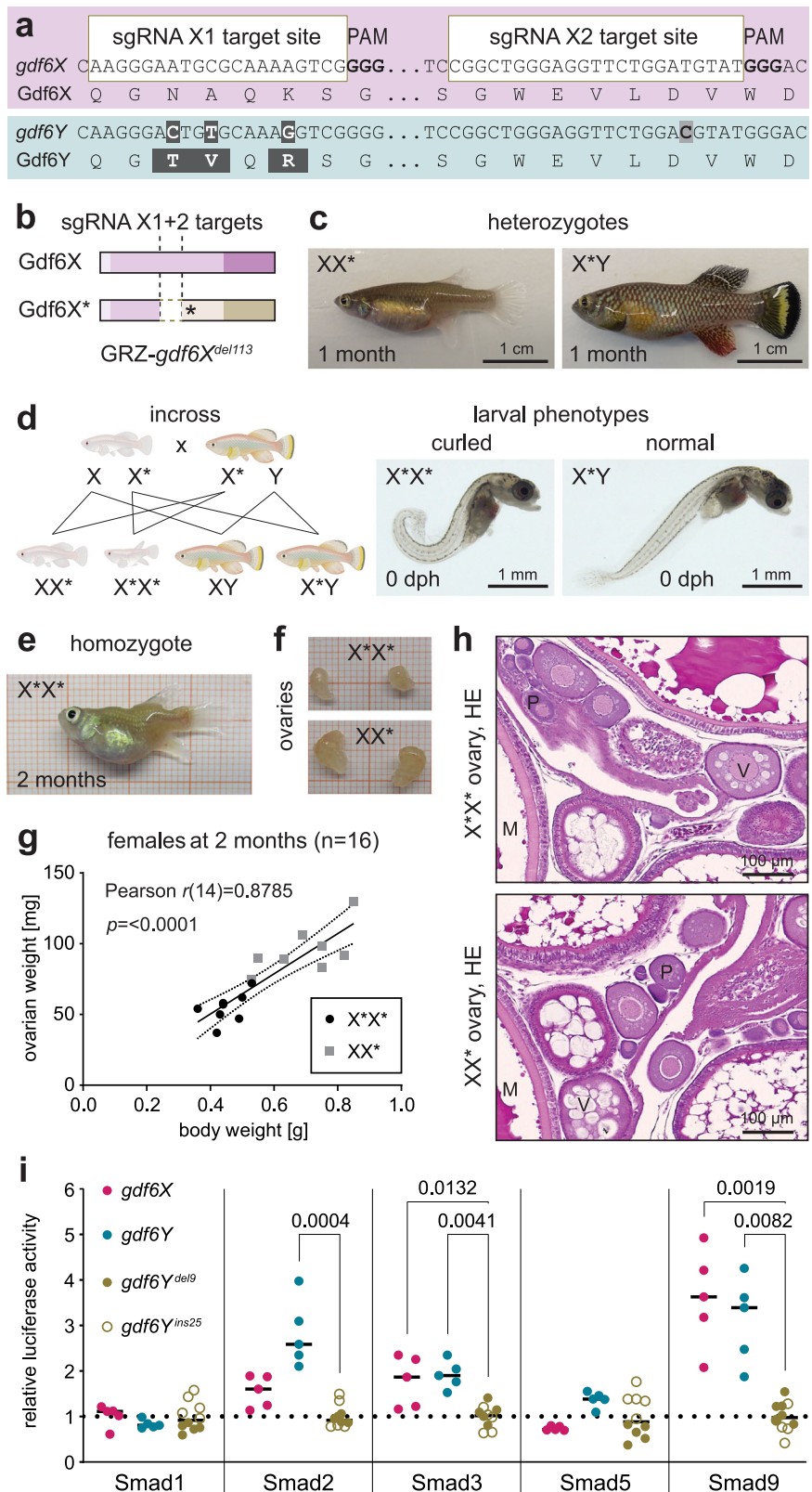

address this question, we employed Cas9 and two sgRNAs targeting *gdf6X* on the X chromosome at positions corresponding to those previously used to inactivate the Y-chromosomal allele *gdf6Y* (Fig. 2a). Using the Synthego ICE (Inference of CRISPR Editing) analysis tool (v3), we estimated an average mutation rate of 97% and 14% in *gdf6X* and *gdf6Y*, respectively, indicating that the inactivation of *gdf6X* was as efficient but not as specific as in the case of *gdf6Y* (Supplementary Fig. 3e).

We observed phenotypes in about one-third of the mosaic F0 X*Y and two-thirds of the mosaic F0 X*X* animals (Supplementary Fig. 3f). Two out of ten mosaic F0 males had one-sided microphthalmia, a probably clonal phenotype that was not passed on to subsequent generations. Furthermore, one X*Y and six out of nine X*X* mosaic F0 animals had a curled tail after hatching and were not raised. This curled phenotype was also found at varying degrees in dechorionated *gdf6X/gdf6Y* double mutant offspring (X*Y*) of XY* phenofemales and

**Fig. 2 | Gdf6Y can functionally cover for Gdf6X.** Schematic of (**a**) the *gdf6X* inactivation and (**b**) the obtained open reading frame disruption of the propagated mutant line GRZ-*gdf6X*[delI13]. **c** Phenotype of heterozygous GRZ-*gdf6X*[delI13] animals, of which both sexes, females (XX*) and males (X*Y), were 1st observed in F2 due to the X-linked heredity of the mutation that was transferred from an F0 father.
**d** Homozygous GRZ-*gdf6X*[delI13] F3 animals (X*X*) were generated by an incross of heterozygous F2 animals and have a curled phenotype after hatching (brightfield microscopy), while siblings (X*X, X*Y, XY) are normal. X*Y is shown for comparison. Schematic created in BioRender. Richter, A. (2024) https://BioRender.com/x35g473. **e** Phenotype of a 2-month-old homozygous GRZ-*gdf6X*[delI13] F3 female (X*X*). **f** Dissected X*X* compared to XX* F3 ovaries at 2 months of age. **g** Pearson

correlation (two-tailed *p*-value) of body and ovarian weight of homozygous (X*X*) and heterozygous (XX*) GRZ-*gdf6X*[delI13] F3 females (*n* = 8 each) at 2 months of age. **h** HE stained X*X* and XX* F3 ovary at 2 months of age. Oocytes: P – previtellogenic, V – vitellogenic, M – mature. **i** Luciferase reporter assay to test the differential activation of Smad1, 2, 3, 5, and 9 upon expression of *gdf6X* or *gdf6Y* compared to *gdf6Y* mutant variants (*gdf6Y*[del9] and *gdf6Y*[ins25]; *n* = 5 each). *Gdf6Y*[del9] has an in-frame 9 bp deletion in the prodomain's CDS, while *gdf6Y*[ins25] harbors a 25 bp insertion at the same position leading to a frameshift and early stop codon. Stratified statistical testing per Smad by Kruskal–Wallis tests and Dunn's multiple comparisons tests with the grouped *gdf6Y* mutant variants. *P*-values < 0.05 are displayed. **g**, **i** Source data are provided as a Source Data file.

mosaic X*Y F0 males at 0 dph (Supplementary Fig. 3g) and in the homozygous quarter of hatchlings (X*X*) from an incross of the isolated mutant line (Fig. 2d; Supplementary Fig. 3h), which harbors a frameshift-causing 113 bp deletion in *gdf6X* leading to an early stop codon (Fig. 2b; Supplementary Table 1). However, this homozygous phenotype was completely absent in female (XX*) but also male (X*Y) heterozygous animals (Fig. 2c), indicating that *gdf6Y*, aside from its SD function, can functionally compensate *gdf6X* inactivation. Thus, the lack of *gdf6X* does not contribute to the phenotype of the YY* embryos, which have one functional *gdf6Y* copy.

Raising the homozygous animals, the curled phenotype develops into a downwards bent tail within three days caused by a spine kink that allows larvae and juveniles relatively normal swimming behavior (Supplementary Fig. 3i). Between one and three weeks after hatching, however, X*X* animals develop additional, overall deforming spine kinks and their swimming behavior becomes impaired, particularly after feeding likely due to a displacement of the swim bladder (Fig. 2e; Supplementary Fig. 3i). This spine phenotype could be caused by hemivertebrae or vertebral fusions, associated with heterozygous *GDF6* mutant or knockdown alleles in humans[12,22], suggesting the conservation of *GDF6*'s function in the killifish. Despite the spine phenotype, the X*X* animals developed into outwardly female fish, which were significantly smaller and lighter than their XX* sisters at two months of age when they were sacrificed to analyze their gonads (Fig. 2e; Supplementary Fig. 3j). In correlation with these body weight differences, X*X* animals had smaller but histologically verified ovaries like XX* females (Fig. 2f–h; Supplementary Fig. 3j). Hence, we assume *gdf6X* does not influence ovarian development.

Given that *gdf6X* and *gdf6Y*, apart from sex determination, seem to be functionally equivalent, we assessed the signaling activities of the respective proteins using a heterologous cellular reporter system[23]. To this end, medaka cells harboring different Gal4-Smad fusion proteins (Smad1, 2, 3, 5, and 9) and a responsive luciferase reporter were co-transfected with expression plasmids for *gdf6X* and *gdf6Y* as well as two *gdf6Y* mutants as controls. Compared to the two *gdf6Y* mutant variants both *gdf6X* and *gdf6Y* showed similar signaling properties by significantly inducing the activation of Smad3, and 9 (Fig. 2i). Notably, Gdf6Y also led to significant Smad2 activation, while the luciferase activity induced by Gdf6X was lower. Hence, enhanced Gdf6Y signaling could have contributed to *gdf6Y*'s neofunctionalization as the SD gene.

**A subset of the testicular somatic supporting cells expresses *gdf6Y***

As *gdf6Y* covers the functions of *gdf6X* and additionally determines the male sex in *N. furzeri*, we were interested in how it might govern sexual development. It was previously shown that *gdf6Y* is expressed in adult testes while *gdf6X* is not expressed in adult gonads[8] (Supplementary Fig. 4a). To address the cellular localization of *gdf6Y*'s testicular expression, we used the very sensitive and specific in situ hybridization technique RNAscope. We applied a *gdf6Y*-specific probe together with probes for the GC marker *ddx4* and one of the somatic cell markers *amh*, *dmrt1*, or *wt1a*[24–26]. While remaining mutant *gdf6Y* mRNA was

mostly detected in expanding, highly *vasa* mRNA-rich oocytes of phenofemales, we found that *gdf6Y* is co-expressed with all three somatic cell markers in proximity to the spermatogonia of the adult testes (here: 3 months), suggesting *gdf6Y*'s expression by pre-Sertoli cells (Fig. 3a; Supplementary Fig. 4b–d). Next, we analyzed the gonadal expression of *gdf6Y* at 0 dph. Marking the GCs with *ddx4* and the somatic cells with *amh*, *dmrt1*, or *wt1a*, we observed *gdf6Y* expression mostly in a subset of the latter in the still largely undifferentiated male gonads as shown by signal proximity analyses (Fig. 3b, c; Supplementary Fig. 4e, f). Taken together, *gdf6Y* is expressed in both the developing and the adult testes by somatic supporting cells.

In contrast to *gdf6Y* and *gdf6X*, their upstream neighbor *sybu* is expressed in oocytes of the developing and adult ovaries (Supplementary Fig. 4h, i), where its mRNA probably localizes to the vegetal pole as it was shown in zebrafish[27]. To analyze DNA methylation as a potential cause of the differential expression of *gdf6X* and *gdf6Y* as well as *sybu* in adult gonads, bisulfite sequencing of the region between *gdf6X/Y* and *sybu* was performed on male, female, and phenofemale gonads (Supplementary Fig. 5a–c). While the identified DNA methylation patterns were highly symmetrical on both DNA strands, we found that neither the promoter of *gdf6X* nor *gdf6Y* was methylated in any of the analyzed sexes (Supplementary Fig. 5b, c). Conversely, the *sybu* promoter was hypomethylated on the X but hypermethylated on the Y chromosome, which translated to a strictly X-chromosomal expression in testes (Supplementary Fig. 5c, d).

In phenofemales, the methylation of the *sybu* promoter on the Y chromosome decreased compared to males (Supplementary Fig. 5c). Hence, *sybu* transcripts from the Y chromosome were also sequence verified in phenofemale ovaries (Supplementary Fig. 5d). An explanation for the altered Y-chromosomal methylation in phenofemales could be demethylation in the maternal germline and the adoption of the methylation pattern from the paternal X chromosome. Passive maternal demethylation and a take-over of the paternal methylome were shown in zebrafish[28]. This implies that the silenced genes of the Y chromosome are reactivated in phenofemales.

The intergenic region between *sybu* and *gdf6X* or *gdf6Y*, which contains two Y-specific transposable elements (TEs; Supplementary Fig. 5a) was strongly methylated at all analyzed positions, suggesting a transcriptional insulation of the two promoters. Notably, a *sybu* transcript variant with an alternative start site (NCBI Reference Sequence: XM_015949384.2) is expressed in *N. furzeri* ovaries but not testes (Supplementary Fig. 5d). In conclusion, upstream methylation does not seem to control the differential testicular expression of *gdf6X* and *gdf6Y*.

**Oogenesis starts before hatching in female turquoise killifish**

To understand the ongoing processes in the developing gonads, we analyzed previously published transcriptome data[8]. Differentially expressed genes (DEGs) between males and females were identified at three developmental stages: the pharyngula stage during organogenesis at 10 dpf, the hatching stage (0 dph), and the larval stage at 3 dph. In the whole embryo at 10 dpf, 3 transcripts were noticeably upregulated in males and continued to do so in all stages (Fig. 4a;

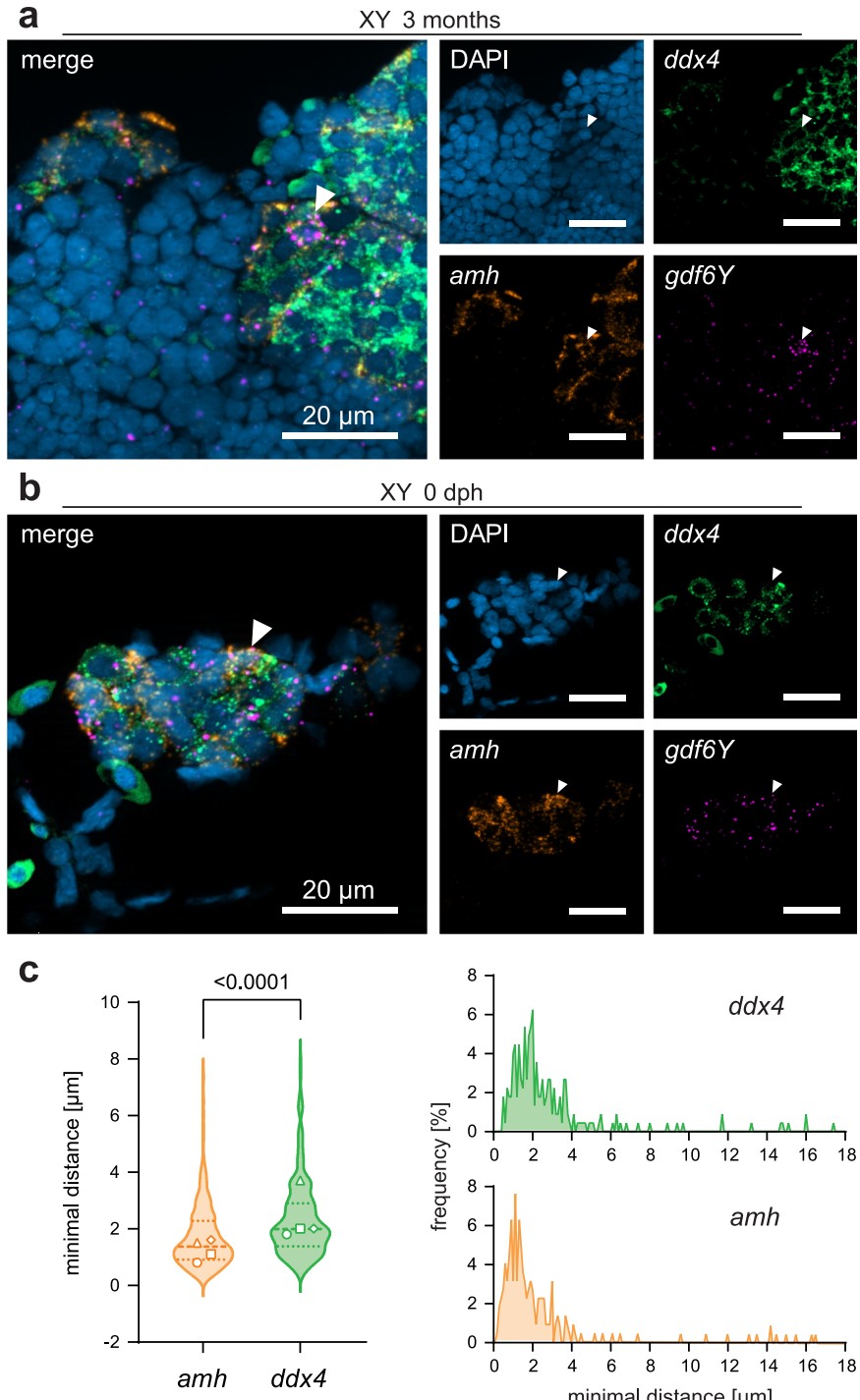

**Fig. 3 | Localization of the gonadal *gdf6Y* expression in XY animals.** Transcripts of *gdf6Y* (magenta) and the GC marker *ddx4* (green) as well as the (**a**) Sertoli or (**b**) somatic supporting cell marker *amh* (orange) were detected in *N. furzeri* testes at (**a**) sexually mature (*n* = 9; here: 3 months old) and (**b**) 0 dph stage (*n* = 7). **a**, **b** White arrows – cells co-expressing *gdf6Y* and *amh*. **c** Minimal distances of *gdf6Y* signals to *ddx4* and *amh* signals from 4 XY gonads at 0 dph including (**b**). Left: Violin plots of minimal distances <8 μm (*n* = 212 per gene, equals lowest 95%) with median, quartiles, and respective modes (bin width: 0.1 μm) from the 4 individual gonads (circle, square, triangle, and diamond). Statistical testing by a two-tailed Mann–Whitney test. Right: Frequency distribution of all measured minimal distances (bin width: 0.1 μm, *n* = 224). Source data are provided as a Source Data file.

Supplementary Data 1). In phenofemales, those transcripts were expressed like in males and, therefore, are not involved in *gdf6Y*-driven sex determination in *N. furzeri* (Supplementary Fig. 6a).

While the number of DEGs increased with age independent of sex, the number of upregulated genes was consistently higher in female embryos (10 dpf) and trunk parts (0 and 3 dph) than in respective male samples (Fig. 4a). From the identified DEGs at 0 dph 53.1% of the

female-specific and 23.3% of the male-specific transcripts were rediscovered at 3 dph (Fig. 4b; Supplementary Fig. 6b). Overrepresentation analyses (ORA) identified enriched KEGG (Kyoto Encyclopedia of Genes and Genomes) pathways for male and female DEGs at 0 and 3 dph. In males, the most enriched pathway was steroid biosynthesis at 0 dph. However, no significantly enriched pathways were identified at 3 dph or within the common DEGs of both stages (Supplementary

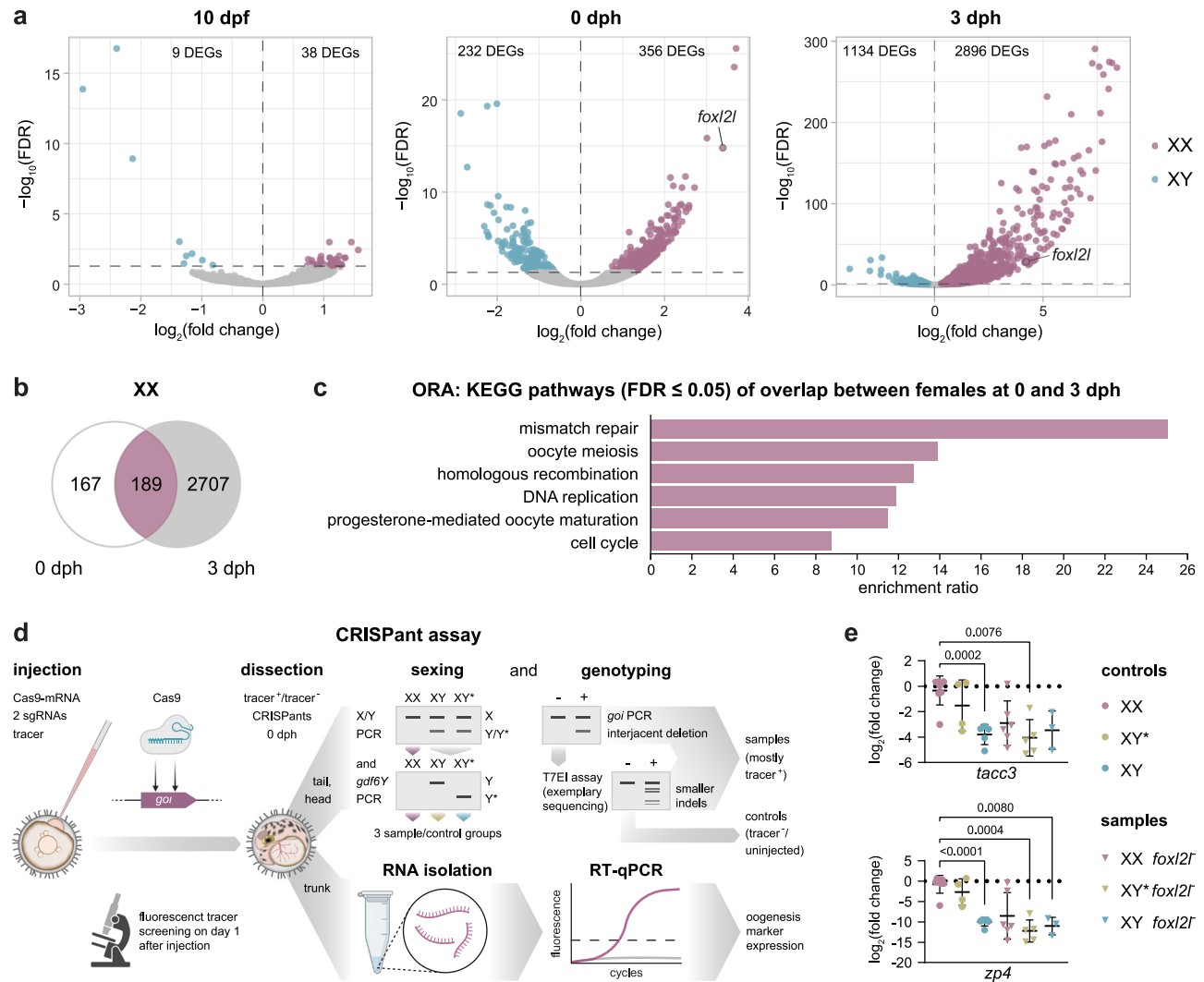

**Fig. 4 | Oogenesis starts before 0 dph, depends on *foxl2l*, and can be measured and intercepted by genetic modification in *N. furzeri*. a** Volcano plots of DEGs (FDR < 0.05) between XX and XY animals in the whole embryo at 10 dpf and in trunk parts at 0 and 3 dph as derived from a respective previously published data set (XX at 0 dph *n* = 2, others *n* = 3; Supplementary Data 1)[8]. **b** Overlap of female DEGs between 0 and 3 dph. **c** Overrepresentation analysis (ORA) of KEGG pathways within the overlap of female DEGs between 0 and 3 dph. **d** Schematic of CRISPR/Cas9-based chimeric mutant (CRISPant) assay to address the influence of a *goi*

(gene of interest) on oogenesis onset as a proxy for sexual differentiation. T7EI – T7 endonuclease I. Created in BioRender. Richter, A. (2024) https://BioRender.com/t02m033. **e** Expression of female marker genes on mRNA-level in trunks of females (XX; *n* = 8), phenofemales (GRZ-*gdf6Y^{del9}* XY*; *n* = 4), and males (XY; *tacc3*, *n* = 5; *zp4*, *n* = 6) and their mosaic *foxl2l⁻* counterparts (XX, *n* = 6; XY*, *n* = 5; XY, *n* = 3) at 0 dph. Welch's ANOVA and Dunnett's T3 multiple comparisons test with XX animals. *P*-values < 0.05 are displayed. **c**, **e** Source data are provided as a Source Data file.

Fig. 6c). In females, pathways related to cell cycle, DNA repair, meiosis, and oocyte maturation were enriched at 0 and 3 dph as well as in the DEG-overlap of both stages (Fig. 4c; Supplementary Fig. 6d, e). This result validates our observation from in situ hybridization experiments that female gonads at 0 dph are larger and more differentiated due to a greater cell number and already developing oocytes (Supplementary Fig. 4g). Therefore, the animals' sex must be determined before 0 dph.

To elucidate this further, we analyzed *gdf6* expression in general and *gdf6Y* expression in particular at 8 different stages of developing embryos between 7 and 14 dpf corresponding to the somite stage, at which facultative diapause occurs, and the completed organogenesis stage, respectively (Supplementary Fig. 6f, g). While *gdf6* expression peaked first directly after leaving the facultative diapause stage and second before the pupil forms during eye development (Supplementary Fig. 6f stages 2 and 6, respectively), no differences were found between the sexes or could be attributed to *gdf6Y* expression (Supplementary Fig. 6g). We also performed whole-mount in situ hybridization of a phenofemale's offspring at 11 dpf (stage 5) with *N. furzeri*

pan-*gdf6* probes allowing for exclusive *gdf6X* and *gdf6Y* detection in XX and YY* embryos, respectively (Supplementary Fig. 6h). In agreement with our finding that *gdf6X* and *gdf6Y* share their developmental role, we observed no differences between the four genotypes (XX, XY, XY*, and YY*), which showed expression of *gdf6X* and/or *gdf6Y* in the ventral half of the body from the rostrum to the end of the yolk-sac extension similarly to the *gdf6* expression in *Danio rerio* at comparable stages[15].

In medaka, *foxl2l* is a GC-intrinsic factor that promotes oogenesis by inducing *rec8a* and suppressing spermatogenesis[19–21]. At 0 and 3 dph, the *N. furzeri* ortholog of *foxl2l* was among the significant DEGs (Fig. 4a). In contrast to the female somatic cell marker *foxl2* and like in medaka, *foxl2l* is upregulated in females at 0 and 3 dph but not in adult gonads (Fig. 1g; Supplementary Fig. 7a, b). Analyzing trunk parts at 0 dph and adult gonads, we showed that *foxl2l* is regulated the same way in phenofemales as it is in females placing the gene's function downstream of *gdf6Y*'s (Supplementary Fig. 7c). Furthermore, RNAscope analyses confirmed a *foxl2l* expression in ovarian GCs at 0 dph by co-localization with *ddx4* (Supplementary Fig. 7d). We also observed *foxl2l*

expression in testicular GCs in proximity to *gdf6Y* expressing somatic cells in one out of six analyzed male individuals at 0 dph (Supplementary Fig. 7e). Searching for *foxl2l* expression foci in adult gonads, we found a single undifferentiated *foxl2l* positive GC in one of three ovaries and none in testes (n = 3; Supplementary Fig. 7f).

To verify that *foxl2l* influences sexual differentiation in *N. furzeri*, we targeted the single exon of *foxl2l* with two sgRNAs simultaneously in males, females, and phenofemales (Fig. 4d). This approach led to various indels disrupting the gene with an efficiency of 63% on average (Supplementary Fig. 7g). Next, we analyzed the expression of oogenesis genes in the trunk parts of the mosaic *foxl2l* mutants at 0 dph. While neither *gdf6X* nor *gdf6Y* expression was changed in the mutants of all sexes, we observed a decrease of marker genes for cell division (*tacc3*) and oocyte development (*zp3/4*, *sybu*) to nearly male levels in the mutant females and phenofemales, which was significant in the latter (Fig. 4e; Supplementary Fig. 7h, i). The aromatase gene *cyp19a1a* and the meiosis marker *rec8*, a putative target of *foxl2l*, decreased only in some but not all mutant females and phenofemales (Supplementary Fig. 7i). Considering the mosaic nature of the mutants, this data supports that *foxl2l* promotes oogenesis in *N. furzeri* and acts downstream of *gdf6Y*.

## Gdf6Y suppresses oogenesis indirectly by determining the male somatic sex

To investigate, if the presence of *gdf6Y* is sufficient to suppress oogenesis in *N. furzeri* development, we used Tol2 transgenesis to drive *gdf6Y* expression under the control of a ubiquitous *actb2* promoter from *D. rerio*[29] (Supplementary Fig. 7j). Analyzing the trunk parts of transgene-positive mosaic animals at 0 dph, we found that the expression of the female-specific genes *zp3/4*, and *sybu* was significantly decreased in transgenic XX animals compared to non-transgenic females, while *rec8* and the aromatase gene *cyp19a1a* decreased only in some but not all transgenic females (Supplementary Fig. 7k). Considering the artificial and mosaic expression of the *gdf6Y*-transgene, this data suggests *gdf6Y*'s sufficiency for *N. furzeri* sex determination. Overall, the *gdf6Y* expression of both XX and XY transgenic animals was much higher than in the non-transgenic males due to the ubiquitous expression (Supplementary Fig. 7l). Interestingly, from the 17 transgene-positive embryos, harvested for this experiment, 16 turned out to be curled up upon dechorionation, although all embryos looked normally developed within the egg. Similarly, when hatching mosaic *gdf6Y* transgenic animals from an independent experiment, we observed that all five transgene-positive hatchlings failed to straighten their body axes within one to two days and were not able to swim normally (Supplementary Fig. 7m). This phenotype was similar but more severe than the homozygous loss-of-function phenotype (Fig. 2d, e; Supplementary Fig. 3g, i), suggesting that tight spatiotemporal regulation of the *gdf6* alleles is necessary for proper development in *N. furzeri*.

As ubiquitous *gdf6Y* expression leads to an adverse phenotype, preventing the fish from being raised, we attempted another approach to allow *gdf6Y* expression under the influence of its natural regulatory elements and obtain a lower transgene copy number than transposase-based Tol2 transgenesis generates. Therefore, we used a previously constructed bacterial artificial chromosome (BAC, GRZ-B-a-208Dg03)[8] that contains a fragment of *N. furzeri*'s Y chromosome including *gdf6Y*, three upstream genes, and a huge part of the downstream gene desert (Fig. 5a). We injected the purified BAC clone, an injection tracer (*EGFP*-mRNA), *Cas9*-mRNA, and two sgRNAs, one targeting the BAC backbone to linearize the circular molecule to trigger integration and the other targeting the second intron of the safe harbor locus *clybl*[30] as the desired integration site with a pre-determined editing efficiency of 81% (n = 14, Sanger sequencing and Synthego ICE analysis tool v3).

Out of 20 raised and analyzed fish, editing of the *clybl* locus was detected with T7EI assay or Sanger sequencing in all tracer-positive

animals including 4 XY males and 7 XX individuals as well as one tracer-negative XX individuals. While the latter and one tracer-positive XX individual developed as females, 6 tracer-positive XX fish showed masculinization over time and had streak gonads (phenomales; Fig. 5b). At 1.5 months of age, 5 out of 6 phenomales had a male body shape and size but female coloration while XY males (n = 7) from the same clutch had all male sex characteristics including full male coloration. Male coloration was first observed in two phenomales at 2.75 months of age. When the phenomales were sacrificed at 4.2 months, 4 were colored and 2 showed no or very pale coloration (Fig. 5b). However, the male-typical operculum decoration was observed in all phenomales. Furthermore, the body length and weight of phenomales resembled those of males and not females of the same age (Fig. 5c, d).

Interestingly, colored but not pale phenomales triggered female egg deposition to the same extent as XY males indicating male breeding behavior (Fig. 5e). None of the collected eggs from phenomale breeding pairs were fertilized as the streak gonads lacked germ cells as verified histologically (Fig. 5f). While in all phenomales the streak gonads were identified as inwardly folded and ciliated tubes, which reminded of the gonadal ducts, a section containing empty tubular structures reminiscent of germ-cell-free testes in other fish species[31,32] was observed in one of the early-coloring phenomales (Fig. 5f). The BAC clone was only detected and sequence confirmed without targeted integration in the fin tissue of this colored and one pale phenomale suggesting high mosaicism and its potentially episomal rather than integrated presence in the fish. This highly mosaic nature of *gdf6Y* in phenomales could explain the belated sex reversal compared to normal male development and the loss of germ cells due to a potentially conflicting sexual differentiation of the germ line and the gonadal soma. However, *gdf6Y*-mosaic phenomales developed somatic male characteristics like XY animals despite lacking germ cells. In contrast, germ-cell-removed *N. furzeri* still develop as males or females depending on their sex chromosomes[33,34]. Hence, we conclude that *gdf6Y* initiates sex determination by inducing the male fate of gonadal somatic supporting cells to pre-Sertoli cells in an auto- or paracrine fashion in the turquoise killifish.

## Transcriptome analysis identifies Gdf6Y-responsive genes

To identify Gdf6Y target genes, we transfected cell lines with a *gdf6Y* expression plasmid or one of two control plasmids and analyzed their transcriptomes after 24 h. As a model for the somatic cells, which express *gdf6Y* in the killifish's testes, we used murine Sertoli-like TM4 cells[35]. The transcriptome of TM4 cells transfected with the *gdf6Y* expression plasmid was compared with the transcriptome of TM4 cells either expressing the *gdf6Y* mutant variant *gdf6Y^{del9}* or carrying an empty vector (EV; Supplementary Data 2). Overlaps of the DEGs from both comparisons resulted in 20 up- and 22 downregulated genes in the presence of *gdf6Y* (Fig. 6a; Supplementary Fig. 8a). An over-representation analysis revealed TGF-β signaling as the most enriched pathway among those 42 DEGs (Fig. 6b). Out of the 7 genes assigned to this pathway, only *Tgfbr2* was downregulated, while *Id1*, *Id2*, *Id3*, *Smad6*, *Smad7*, and *Smad9* were upregulated (Fig. 6a). With *Id1*, *Id2*, and *Id3* being known downstream effectors of BMPs[36], this suggests that these genes are regulated as an immediate response to Gdf6Y signaling.

To verify the identified Gdf6Y-responsive genes, we performed the same experiment in human HeLa cells (Fig. 6c; Supplementary Fig. 8b; Supplementary Data 3). Here, 6 commonly upregulated orthologous genes were identified, namely *LXN*, *ID1*, *ID2*, *ID3*, and the inhibitory Smad-genes *SMAD6* and *SMAD7*, of which the latter 5 were previously assigned to the TGF-β signaling pathway (Fig. 6a). The single commonly downregulated orthologous gene was *ZFP36L2*, which encodes a zinc finger protein that binds to AU-rich elements (ARE) in the 3′ untranslated regions of mRNAs promoting their degradation[37].

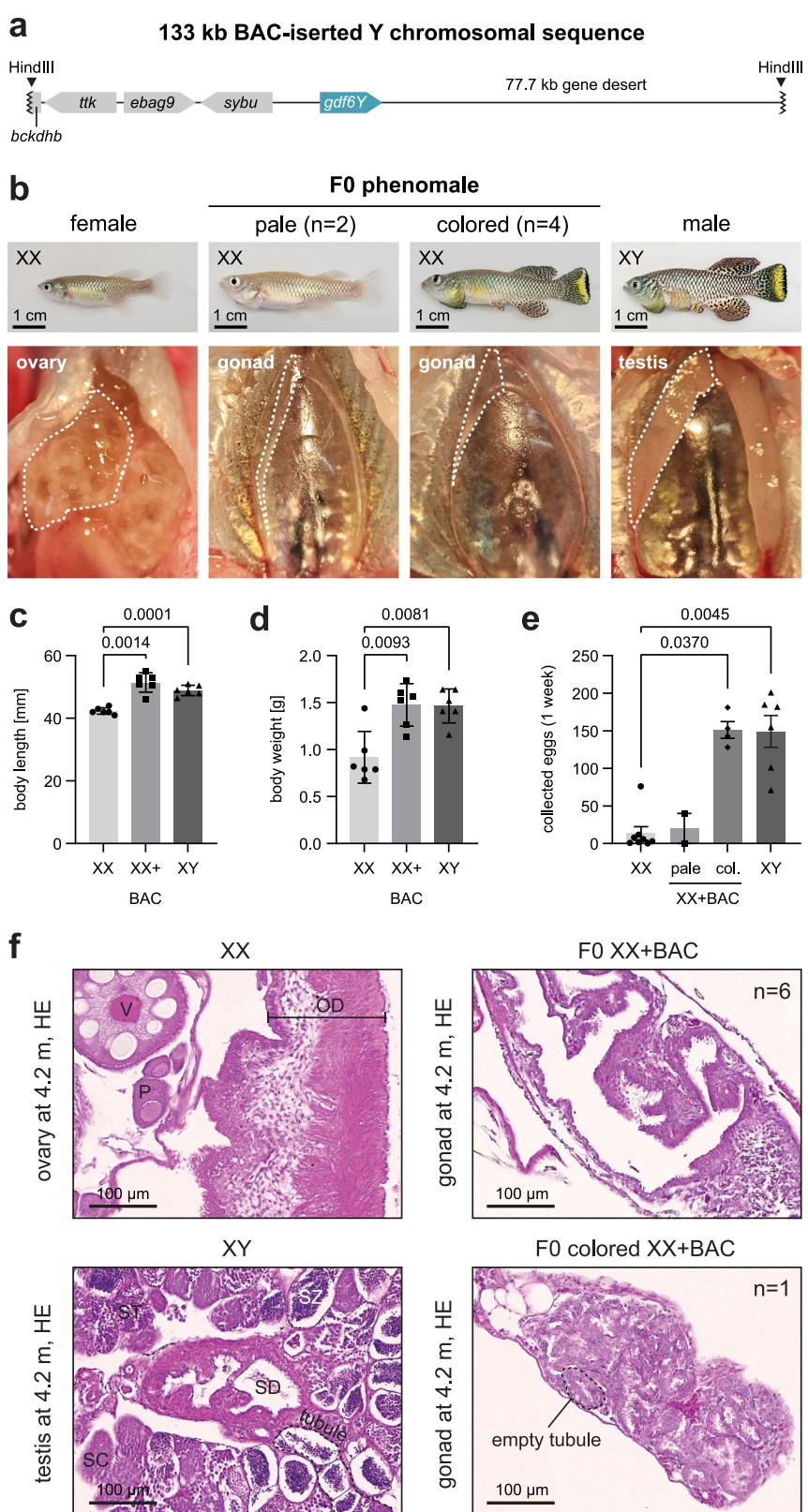

**Fig. 5 | A *gdf6Y*-containing Y-chromosomal fragment leads to a female-to-male sex reversal. a** Schematic of the Y-chromosomal sequence inserted via HindIII into the bacterial artificial chromosome (BAC) GRZ-B-a-208Dg03[8]. **b** Phenotypes and streak gonads in situ of partially sex-reversed mosaic XX animals (phenomales) created by zygotic BAC injection compared to an XX female and an XY male at 4.2 months of age. Body length (**c**) and body weight (**d**) of phenomales (XX + BAC) compared to females and males at 4.2 months of age (*n* = 6 each). Statistical testing by Welch's ANOVA and Dunnett's T3 multiple comparisons test. **e** Numbers of all eggs (dead or alive) collected after 1 week of constant breeding of a single female (2.5 months) with another female (XX, 2.5-5 months, *n* = 8), a pale (4 months, *n* = 2) or a colored phenomale (XX + BAC, 4 months, *n* = 4), or a male (XY, 4.2 months, *n* = 6). Statistical testing by a Kruskal–Wallis test and Dunn's multiple comparisons test with XX. **c**–**e** Mean with standard deviation. *P*-values < 0.05 are displayed. Source data are provided as a Source Data file. **f** HE stained gonadal sections of two BAC-positive individuals with the observation frequency in all phenomales. Ovary and testis connecting to the respective gonadal duct are given for comparison. Oocytes: P – previtellogenic, V – vitellogenic; OD - oviduct. SC – spermatocytes, ST – spermatids, SZ – spermatozoa; SD – sperm duct.

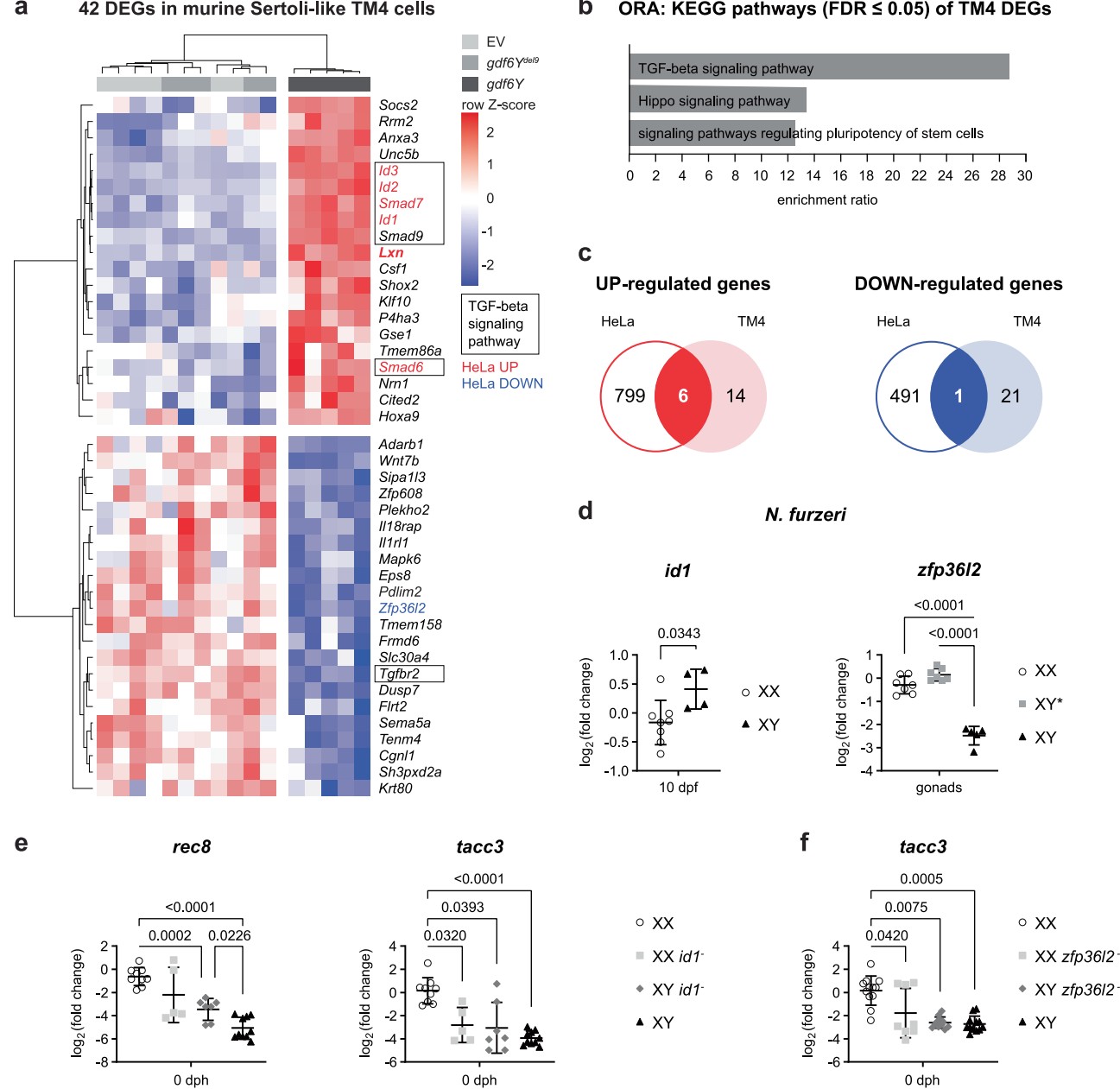

**Fig. 6 | Gdf6Y leads to the specific regulation of certain genes independent of the targeted cell's identity. a** Heatmap of DEGs (FDR < 0.05) between murine Sertoli-like TM4 cells transfected with a *gdf6Y* expression plasmid (*n* = 6) and TM4 cells transfected with either a *gdf6Y* mutant variant (*gdf6Y*^*del9*^, *n* = 5) expression plasmid or an empty vector (EV, *n* = 6). Genes of the TGF-beta signaling pathway are boxed. **b** Overrepresentation analysis (ORA) of KEGG pathways within the 42 DEGs identified in TM4 cells. FDR − false discovery rate. **c** Overlap of the up- or down-regulated DEGs identified in TM4 cells with those identified in human HeLa cells. The common genes are highlighted in red (upregulated genes) or blue (the downregulated gene) in (**a**). **d** RT-qPCR analyses of the expression of *id1* and *zfp36l2* in female (XX, *n* = 8) and male (XY, *n* = 4) embryos at 10 dpf (stage 4) or gonads of

3 months old females (XX, *n* = 7), males (XY, *n* = 5), and phenofemales (GRZ-*gdf6Y*^*del9*^ XY*, *n* = 7), respectively. Two-tailed Welch's *t*-test (*id1*) or Welch's ANOVA and Dunnett's T3 multiple comparisons test (*zfp36l2*). **e** RT-qPCR analyses of female marker genes in trunks of uninjected (*n* = 6 each) and indel-negative (XX, *n* = 3; XY, *n* = 4) females and males and *id1*^−^ CRISPants of both sexes (XX, *n* = 5; XY, *n* = 7) at 0 dph. Welch's ANOVA and Dunnett's T3 multiple comparisons test. **f** RT-qPCR analyses of female marker genes in trunks of uninjected (*n* = 9 each) and indel-negative (*n* = 3 each) females and males and *zfp36l2*^−^ CRISPants of both sexes (XX, *n* = 8; XY, *n* = 11) at 0 dph. Kruskal–Wallis test and Dunn's multiple comparisons test. **d**–**f** Mean with standard deviation. *P*-values < 0.05 are displayed. **a**, **b**, **d**–**f** Source data are provided as a Source Data file.

Expression analysis in accordingly treated human HEK293 cells confirmed the regulation of all these orthologous genes but *ID2* in the presence of *gdf6Y*, verifying them as cell-type independent Gdf6Y-response genes (Supplementary Fig. 8c).

Next, we assessed the sex-specific expression of the *N. furzeri* orthologs in different tissues and developmental stages utilizing the previously published transcriptome data from Reichwald et al.[8]. While no reads had been mapped for the orthologs of *Id2* and *Lxn*, we found

that *id1, smad6b,* and *zfp36l2* were significantly up- or downregulated in males at 3 dph, respectively, corresponding to the regulation direction observed in cell culture (Supplementary Fig. 8d). The male-specific upregulation of *id1* was experimentally confirmed at 10 dpf (stage 4 in Supplementary Fig. 6f), marking it the earliest timepoint of differential regulation of a Gdf6Y-responsive gene in *N. furzeri* males and females (Fig. 6d). The male-specific downregulation of *zfp36l2* was furthermore reversed in phenofemales' gonads carrying either the

*gdf6Y* mutant variant *gdf6Y*<sup>del9</sup> (Fig. 6d) or the frameshift mutation *gdf6Y*<sup>del113</sup> (Supplementary Fig. 8e). This also applied to the male-specific upregulation of *smad9*, which is the Smad-gene exclusively identified in Sertoli-like TM4 cells (Supplementary Fig. 8e).

To further address the potential role of *id1*, *smad9*, and *zfp36l2* in *N. furzeri*'s sex determination, we analyzed their gonadal localization in males and females at 0 dph and in gonads with RNAscope (Supplementary Fig. 9a). At 0 dph, the expression of *id1* was detected in gonadal somatic cells of both sexes and occasionally in female germ cells. In mature gonads, Sertoli and granulosa cells as well as oocytes expressed *id1*. In contrast, *smad9* localized mostly to germ cells of both sexes at 0 dph but was largely absent from germ cells in mature ovaries and testes, where it was instead detected in Sertoli cells. As known from the literature[38,39], oocytes strongly expressed *zfp36l2* in both analyzed stages. In addition, *zfp36l2* was also found in granulosa cells of the mature ovary. In male gonads at 0 dph, little *zfp36l2*-signal was detected, and in 2 out of 3 samples in *amh*-negative somatic cells. In mature testes, *zfp36l2*-positive germ and somatic cells were few and scattered.

As *smad9* seemed to be a feature of determined Sertoli cells, we focused on *id1* and *zfp36l2*, which were localized to somatic cells at 0 dph. We performed the above-described CRISPant assay for both (Fig. 4d), expecting male-to-female and female-to-male sex reversal for *id1* and *zfp36l2*, respectively, in case either is involved in sex determination. Therefore, we targeted *id1*'s first exon with two sgRNAs simultaneously. This approach led to various indels disrupting the gene with a varying efficiency of 23–90%. Analyzing the expression of oogenesis marker genes in the trunk parts of XX and XY mosaic *id1* mutants at 0 dph, we found a significant increase of the meiosis marker *rec8* in XY CRISPants compared to control males (XY), however, not to the level of the control females suggesting a male-to-female sex-reversing effect (XX; Fig. 6e). We also observed a decrease in the expression of all analyzed oogenesis marker genes but *cyp19a1a* in *id1* CRISPant females compared to female controls, which only reached significance for *tacc3* (Fig. 6e; Supplementary Fig. 9b). As *id1* is expressed in both male and female gonads of the analyzed stage (Supplementary Fig. 9a) and is known to be involved in human fetal germ-cell development in both sexes as well[40], *id1* inactivation might interfere with male sex determination first and female germ-cell development later, leading to the observed results.

For *zfp36l2*, two sgRNAs simultaneously targeting its second exon led to an average mutation rate of 72%. While only reaching significance for *tacc3*, the expression of all analyzed oogenesis marker genes was reduced in some but not all XX *zfp36l2* CRISPants compared to the female controls (Fig. 6f; Supplementary Fig. 9c). Hence, we observed the expected female-to-male sex-reversing effect. As the readout of our CRISPant assay is oogenesis-based and *zfp36l2* also plays a role in oocyte maturation[38,39], its role in sexual development needs further investigation. However, our data suggest that the identified Gdf6Y target genes *id1* and *zfp36l2* are involved in *N. furzeri* sex determination.

## Discussion

Employing CRISPR/Cas9-mediated *gdf6Y* inactivation and *gdf6Y*-expression strategies resulting in XY male-to-female and XX female-to-male sex reversals, respectively, we demonstrate that *gdf6Y* is necessary and sufficient for sex determination in *N. furzeri*. Among the identified *gdf6Y* mutations causing a male-to-female sex reversal, we found a 9 bp in-frame deletion within the prodomain (*gdf6Y*<sup>del9</sup>), which leads to the loss of amino acids 199–201. These amino acids are highly conserved among vertebrate *gdf6* orthologs and a heterozygous point mutation altering one of them leads to microphthalmia in humans[12]. While not directly affecting the mature ligand, these residues might be important for prodomain folding and the release of the ligand from the proprotein complex, which is required for signaling. All other *gdf6Y*

mutants harbored frameshift mutations leading to premature translational termination in the latter half of the prodomain. Similar frameshift mutations in *gdf6b* also led to male-to-female sex reversal in the Pachón cavefish *Astyanax mexicanus*[41], which harbors two *gdf6* paralogs, *gdf6a*, and *gdf6b*. While the SD gene in *A. mexicanus* arose from an additional *gdf6b* duplicate on a supernumerary B chromosome, *gdf6Y* and conversely *gdf6X* emerged through allelic diversification from an ancient *gdf6a* as no *gdf6b* ortholog is present in *N. furzeri*'s genome sequences[8,9].

While *gdf6Y* compensates for the loss of *gdf6X*, it also adopted the SD function in *N. furzeri* and its sister species *N. kadleci* about 2 million years ago[42]. Three possible scenarios of how *gdf6Y*'s SD function might have evolved comprise functional alterations, changes in the spatio-temporal expression pattern, or a combination of both. The positive selection of *gdf6Y* in the SDR and the conservation of non-synonymous SNVs among strains support the theory of an allelic neofunctionalization[8]. Structural modeling suggested that the four most C-terminal amino acid substitutions in Gdf6Y affect dimerization or ligand-receptor binding[8]. We found that Smad2 was significantly activated by Gdf6Y but not by Gdf6X. Interestingly, the substitution of a highly conserved tyrosine for asparagine in human GDF6 decreased its sensitivity for the inhibitor Noggin[43]. In *N. furzeri*, this residue is surrounded by the two most C-terminal amino acid substitutions between Gdf6X and Gdf6Y, suggesting decreased inhibitor sensitivity as an explanation for Gdf6Y's enhanced signaling capability.

On the other hand, we confirmed the expression of *gdf6Y* but not *gdf6X* in adult *N. furzeri* testes, implying a differential regulation of the alleles. We investigated DNA methylation as a potential cause of this phenomenon. While X-chromosomal hypermethylation of the SDR is crucial for sex determination in the channel catfish *Ictalurus punctatus*[44], the promoters of both *gdf6Y* and *gdf6X* are hypomethylated in gonads. In contrast, the Y-chromosomal *sybu* promoter is hypermethylated indicating transcriptional silencing of *N. furzeri*'s Y chromosome outside of the SD locus. TEs in other fish species contribute sex-specific positive and negative regulatory elements crucial for SD gene function[23,45]. However, the intergenic region between *gdf6X*/*gdf6Y* and *sybu* was highly methylated on both sex chromosomes. Therefore, the two Y-exclusive TEs in this region might not harbor *gdf6Y*-specific upstream enhancers, as enhancer activity would require a low or at most intermediate DNA methylation[46].

Given that TEs accumulate on sex chromosomes during their early evolution[47] and contribute to the suppression of meiotic recombination[48], they could also cause an aberrant expression of non-SD genes in the SDR e.g., the Y-dependent expression of *pctp* in *N. furzeri* embryos. However, the Y-chromosomal effects on the expression of *rnf19a* and *pctp* might be GRZ-specific, as the SDRs of *N. furzeri* strains strongly differ in size with the smallest containing only one gene, namely *gdf6Y* itself[8]. Even allelic SNVs in *sybu* are not sex-linked in all strains[8]. Therefore, the severe phenotype of YY* embryos might not occur in strains with small SDRs.

Conversely, regulatory elements, causing the testis-specific expression of *gdf6Y*, should be located within the minimal SDR, which also contains a gene desert downstream of the gene itself. This well-conserved gene desert harbors *cis*-regulatory elements controlling *GDF6* expression in other vertebrates[15,49,50]. Therefore, changes in or the acquisition of new Y-chromosomal enhancers downstream of *gdf6Y* are potential causes of its sex-specific expression. As our BAC-transgenesis approach led to an XX female-to-male sex reversal, this potential enhancer should be located within the 77.7 kb portion of the gene desert included in the BAC's sequence. Whether protein or expression changes evolved first, and which event constituted *gdf6Y* as the SD gene are open questions, but both probably played a role in its fixation.

Vertebrate sex determination is usually initiated by the sexual fate decision of the somatic cells of the bipotential gonadal primordium,

which in turn control female or male gametogenesis. In a trade-off between male and female regulatory networks involving *Dmrt1* on the male and *Foxl2* on the female side, the equilibrium tips to one side by varying SD factors in different species[51]. In *N. furzeri*, *gdf6Y* is the male SD gene expressed in testicular somatic cells, where *gdf6Y* also induces the male sex of the organism. In a cell culture-based approach murine Sertoli-like cells were used to identify Gdf6Y targets. In a complementary approach, Wang et al. [52] analyzed gene expression in response to BMP signaling inhibition in human testis cells from 7-week-old fetuses, which is one to two weeks past the arrival of the migrating GCs in the genital ridge and, thereby, shortly after sex determination. As in our data, *ID1/2/3*, *SMAD6/7/9*, and *UNC5B* were upregulated in a BMP-dependent manner. However, only *UNC5B* was Sertoli cell-specific, while the other genes were regulated in all somatic and GC populations, showing the omnipresence of BMP signaling in the developing testis. This is consistent with our observation that *Id1/2/3* and *Smad6/7* are expressed in response to Gdf6Y independent of cell type and the finding of *id1*-transcripts in germ and somatic supporting cells of not only *N. furzeri* testes but also ovaries, indicating multiple *id1* functions in *N. furzeri* gonads.

*Arapaima gigas* provided evidence of the involvement of *ID* genes in sex determination in teleost species with a Y-chromosomal *id2b* duplicate being its SD gene[53]. In *N. furzeri*, *id3* is positively selected like *gdf6Y*, suggesting a potential co-evolution[8]. However, we found *id1* upregulated in male *N. furzeri* embryos at 10 dpf and a significant increase in the expression of the meiotic marker gene *rec8* in males at 0 dph by *id1* inactivation, suggesting that *id1* might be the key player among the *id* genes responding to Gdf6Y and that sex determination in *N. furzeri* starts around or before 10 days post-fertilization.

In mammals, *N. furzeri*, medaka, and zebrafish, female GCs enter meiosis during early gonadal development, while male GCs do not until puberty. Regarding the determination of the GCs' sexual fate, *foxl2l* was identified as a GC-intrinsic feminizing factor in medaka, where the gene's inactivation leads to a GC-specific female-to-male sex reversal and the development of sperm in ovaries that failed to initiate oogenesis at the regular developmental time point[19]. While *foxl2l* inactivation in tilapia (*Oreochromis niloticus*) leads to a similar phenotype, it leads to a full female-to-male sex reversal in zebrafish[54,55].

In *N. furzeri*, *foxl2l* is also involved in female GC sex determination, as shown by the decreased oogenesis marker gene expression in mosaic *foxl2l* inactivated females and phenofemales at 0 dph. In medaka, male and female GCs initially express *foxl2l* at the onset of sexual differentiation of the gonads. While its expression is lost in male GCs, the mitotically active subpopulation of female GCs expresses *foxl2l* until entering meiosis throughout life[19,56,57]. The role of *foxl2l*-expressing cells as pre-meiotic oocyte progenitors was confirmed in the zebrafish ovary[55]. In *N. furzeri*, we also detected *foxl2l* expression in female GCs but not oocytes during ovarian development at 0 dph and adulthood. In contrast, we rarely found *foxl2l*-expressing GCs in developing testes at 0 dph and never in adult testes, supporting that its female germ-cell-specific SD function is conserved in *N. furzeri*.

Interestingly, GC ablation leads to the development of the male sex in the small laboratory fish species *D. rerio*[58] and the medaka, *Oryzias latipes*[59], indicating a somatic male predisposition and a GC-dependence of the female trajectory. In contrast and like in mammals[60], male and female development is GC-independent in other teleost species, including the loach (*Misgurnus anguillicaudatus*)[61], goldfish (*Carassius auratus*)[31], salmon (*Salmo salar*)[32], and as recently discovered in the model organism *N. furzeri*[33,34]. Apart from *gdf6Y*, two male SD genes from the previously mentioned teleost species are known, namely *DMY/DMRT1Y*[62,63] in the medaka and *sdY* in salmonids[64,65], hence, from one species with and another without GC-dependent female sex determination, respectively.

Similar to our observation that turquoise killifish ovaries have a higher GC content than testes at 0 dph, the first sign of sexual differentiation in the medaka is the increase in primordial GC (PGC) number in females compared to males[26]. As *DMY* is expressed by the somatic supporting cells accompanying the PGCs[26], it was hypothesized that *DMY* enables future Sertoli cells to regulate PGC proliferation[26,66] and thereby, in a way suppresses female development. While this theory fits well with the GC-dependence of the female sex in the medaka, it is unlikely in the turquoise killifish, as *gdf6Y* is indispensable for male characteristics even in the absence of GCs[33,34]. Hence, *gdf6Y* rather determines the male sex similarly to the salmonid SD gene *sdY*, which suppresses female development on the level of the somatic supporting cells by directly interacting with Foxl2 to suppress aromatase gene expression as shown in *O. mykiss*[67]. However, we found neither *FOXL2* nor *DMRT1* or *SOX9* homologs among the genes significantly regulated in response to Gdf6Y (Supplementary Data 2 and 3) pointing towards a more indirect suppression of the female or enhancement of the male pathway, respectively, in the gonadal somatic cells.

Our data suggest that Gdf6Y triggers a potentially autocrine TGF-β signaling response in the male gonadal somatic supporting cells. This response is characterized by the induction of *Id* genes, which encode inhibitors of DNA binding that influence transcription by preventing the DNA binding of basic helix loop helix transcription factors[68]. Therefore, Gdf6Y could suppress female or enhance male differentiation of the gonadal soma through *id* gene induction leading to global transcriptional changes.

We also identified one common downregulated gene in response to Gdf6Y, namely *zfp36l2*, which encodes a tristetraprolin (TTP)-like mRNA decay activator. In HeLa cells, not only *ZFP36L2* but also the other TTP-family members *ZFP36* and *ZFP36L1* were significantly downregulated in response to Gdf6Y, indicating them as potential GDF6 signaling targets. *Zfp36l2* is expressed in developing oocytes, where it is involved in meiotic cell cycle progression and oocyte maturation[38,39,69], which our localization and expression data confirm for *N. furzeri* ovaries.

While this takes place after sex determination, *Zfp36l2* was also shown to be expressed in porcine granulosa cells, where its expression was inversely correlated with the expression of *Id2/3*[70]. This is consistent with the observed granulosa cell localization of *zfp36l2*-mRNA in killifish ovaries and the directions of *Zfp36l2* and *Id* regulation in response to Gdf6Y in our cell culture approach, suggesting a negative regulation of *Zfp36l2* by *Id* gene activity. Although *zfp36l2* and *id1* have a similar cellular localization in mature gonads, the *zfp36l2* expression in *amh*-negative somatic cells and the *id1* expression in *amh*-positive, hence, committed somatic supporting cells in XY gonads at 0 dph supports this theory. Notably in this context, *id1* was also upregulated in males at 10 dpf before female sexual development was confirmed. The female-to-male sex-reversing effect, we observed upon *zfp36l2* inactivation furthermore indicates that *zfp36l2* plays a role in *N. furzeri* female sexual differentiation, which could be suppressed by the Gdf6Y-signaling response involving *id* genes. Zfp36l2 also controls epigenetic changes in murine oocytes and bone by regulating transcripts of histone modifiers and DNA demethylases[39,71]. This indicates that Gdf6Y signaling might influence the epigenetic landscape of cells by *zfp36l2* repression and thereby determine their sexual fate.

## Methods
### Fish experiments
All animals were maintained in the fish facility of the Leibniz Institute on Aging – Fritz Lipmann Institute Jena according to the German Animal Welfare Law. All experiments were covered by animal experiment licenses 03-005/12, 03-044/16, FLI-18-020, and FLI-21-012 approved by the Thuringian authorities (Thüringer Landesamt für Verbraucherschutz).

Experiments were performed in both sexes of the *N. furzeri* wild-type strain GRZ[2] or genetically modified lines in its background

generated in this study (Supplementary Table 1) except for the in situ hybridization in Supplementary Fig. 4i, where the strain MZM0403 with the smallest SDR but the same SD mechanism as other *N. furzeri* strains and the sister species *N. kadleci* was used instead[3,8,42]. Fish were single-housed at 26 °C in a light:dark cycle of 12 h each. Juveniles up to 5 weeks post-hatching were fed twice daily with artemia, while adult fish, as we refer to them at 5 weeks and older, were fed *ad libitum* once daily with red mosquito larvae. To obtain fertilized GRZ oocytes for injections, breeding groups of five (one male, four females) to ten fish (two males, eight females) were set up in 20–40 l tanks, respectively. To obtain zygotes from phenofemales, breeding groups of one male and up to four phenofemales were set up in 8.5 l tanks. Sandboxes for egg deposition were removed for two days and placed back into the tank 2 h before egg collection with a sieve and subsequent micro-injection. For line maintenance and fertility assessment, eggs were collected weekly, put on coconut coir plates, and incubated at 29 °C to develop until they were ready to hatch. For egg storage up to 1 year, eggs were incubated at 25 °C to enter diapause and moved to 29 °C for 1–2 weeks before hatching.

Phenotypes of embryos, larvae, and juveniles were documented using reflective or transmission light with the Zeiss SteREO Discovery.V8 equipped with a Zeiss AxioCam MRc and a Volpi dual goo-seneck pole mount light or the Zeiss Axio Zoom.V16 fitted with a Zeiss AxioCam HRc and an HXP 200C light source, respectively. Phenotypes of adult animals were documented with Canon IXUS 160.

## Genomic modifications using CRISPR/Cas9

To generate mutant or transgenic animals with CRISPR/Cas9, sgRNAs were designed based on published X- and Y-chromosomal sequences[8] (*gdf6Y*- and *gdf6X* inactivation) or by utilizing CHOPCHOP v3[72] (*foxl2l*, *id1*, *zfp36l2* inactivation; BAC-linearization; *clybl*-opening). Templates for the in vitro transcription of sgRNAs were generated either by subcloning annealed oligonucleotides containing the target sequences into pDR274 (Addgene, plasmid #42250) via *BsaI* (*gdf6Y* inactivation) or by PCR (others). For the PCR, complementary oligonucleotides containing the T7 promoter, the 20 bp sgRNA target sequence, and the sgRNA backbone were applied as template-DNA at a concentration of 0.002 μM each and amplified using Phusion High-Fidelity Polymerase in 1× HF-Buffer (Thermo Fisher Scientific) with an additional primer pair (T7F and pDRrvs) according to the manufacturer's instructions. The respective oligonucleotides are listed in Supplementary Table 2. The approximately 300 bp *DraI* restriction fragments (*gdf6Y* inactivation) or the 124 bp amplicons (others) were purified from an agarose gel or the PCR, respectively, using the NucleoSpin Gel and PCR Clean-up Mini kit (Macherey-Nagel) according to the manufacturer's instructions. Up to 1 μg purified DNA was in vitro transcribed overnight at 37 °C with the MAXIscript T7 Transcription Kit (Thermo Fisher Scientific). The synthesized sgRNAs were precipitated from the reaction using Ammonium acetate Stop Solution as described for the mMessage mMachine Kit protocol (Thermo Fisher Scientific).

*Cas9*-mRNA was in vitro transcribed overnight at 20 °C from 400 to 500 ng *XbaI*-linearized pT3TS-nCas9n (Addgene, plasmid #46757) using the mMessage mMachine T3 Kit (Thermo Fisher Scientific) and purified with the MegaClear Kit (Thermo Fisher Scientific) according to the manufacturer's instructions. All RNAs were quality-controlled by RNA agarose gel electrophoresis. Injection solutions contained 30 ng/μl of each sgRNA, 300 ng/μl of *Cas9*-mRNA, 30 ng/μl of *EGFP*-mRNA[73], 0.1% phenol red, and facultatively 20 ng/μl BAC GRZ-B-a-208Dg03[8], purified from 850 ml shaking overnight liquid culture at 37 °C with QIAGEN Plasmid Maxi Kit adapted for Very Low-Copy Plasmid/Cosmid Purification Using QIAGEN-tip 500 according to the QIAGEN Plasmid Purification Handbook.

*N. furzeri* zygotes were placed in slots of a 1.5% agarose plate in 0.3× Danieau's medium prepared with a mold[74] (GT-Labortechnik, Arnstein, Germany) and microinjected under a stereomicroscope (Olympus SZ61 with KL 1500 compact light source) with glass capillary (GC100F-10, Harvard Apparatus) needles[73], a micromanipulator (Saur), and a pressure injector (World Precision Instruments). Per batch of collected eggs, up to 324 zygotes were injected. For detailed numbers concerning F0-only experiments see Supplementary Table 3.

## Molecular sexing and genotyping of animals

Genomic DNA was extracted from caudal fin biopsies of adult animals[74] or tail tips from dechorionated embryos and larvae[75] by lysis in 100 μl of 50 mM NaOH for 45 min at 95 °C and subsequent buffering with 10 μl of 1 M Tris-HCl (pH 8.0). For sexing and genotyping PCRs, DreamTaq DNA Polymerase (Thermo Fisher Scientific) was used according to the manufacturer's instructions with the oligonucleotides and annealing temperatures given in Supplementary Table 4. Ethidium bromide-stained agarose gel pictures were acquired using the Bio-Rad Molecular Imager Gel Doc XR+ and Image Lab 6.0.1. Sex chromosomes were determined by either a *gdf6Y*-specific PCR (gonosome and *gdf6Y* PCR in Supplementary Fig. 1b and Fig. 4d, respectively), or sexing PCR amplifying X/Y-length polymorphisms[75] in- or outside of the BAC-cloned Y-chromosomal sequence (Supplementary Fig. 3a and the X/Y PCR in Fig. 4d).

As the *gdf6Y*-specific PCR flanks the target sites of sgRNA Y1 and Y2, it was also used for genotyping of GRZ-*gdf6Y*<sup>del6, del8</sup> and GRZ-*gdf6Y*<sup>del9</sup> phenofemales yielding 205 or 210 bp fragment, respectively, compared to a 219 bp fragment in males. GRZ-g*df6Y*<sup>del113</sup> and GRZ-*gdf6X*<sup>del113</sup> lines were genotyped by a smaller fragment indicating the mutant allele in a pan-*gdf6*-PCR amplifying the target regions on both sex chromosomes (Target locus PCR in Supplementary Fig. 1b, c). Mutations in *foxl2l*, *clybl*, *id1*, and *zfp36l2* were analyzed using oligo-nucleotides flanking the sgRNA target sites. Tol2-generated *gdf6Y*-transgenic hatchlings and larvae were identified by detecting EGFP in the heart by fluorescence microscopy and a transgene-specific 735 bp fragment by applying a Tol2-transgene-specific PCR. A BAC-specific PCR was used to identify the BAC transgene.

## Immunohistochemistry and RNAscope in situ hybridization

Dissected gonads and embryos were fixed in 4% paraformaldehyde/PBS overnight at 4 °C, washed in phosphate-buffered saline, and paraffin-embedded. 5 μm sections were prepared with the Epredia HM 340 E electronic rotation microtome (Thermo Fisher Scientific) and transferred onto Superfrost Plus Slides (Thermo Fisher Scientific). The prepared sections were stained with Haematoxylin and Eosin (H&E) or used for RNAscope experiments. H&E stainings were imaged and processed with a Plan-Apochromat 10×/0.45 M27 objective at the AxioScan.Z1 (Zeiss) and ZEN 2.6 (Blue Edition, Zeiss), respectively.

For RNAscope experiments, if not stated otherwise, Advanced Cell Diagnostics provided buffers, equipment, and probes (Supplementary Table 5). The experiments were performed according to the manufacturer's protocol (Advanced Cell Diagnostics, Document number: 323100-USM) except for the following changes: 60 °C drying steps at the end of deparaffinization and target retrieval were prolonged from 5 to 30 min. Target retrieval and RNAscope Protease Plus treatment were performed for 15 and 30 min, respectively. After creating a barrier with the ImmEdge hydrophobic barrier pen, specimens were photobleached at 350–425 nm for 5 min with Axio Zoom.V16 or the SteREO Discovery.V8 (Zeiss) to reduce auto-fluorescence. Opal 520 (FP1487001KT), 570 (FP1488001KT), and 650 (FP1496001KT) from Akoya Biosciences were used as dyes for C1 (*ddx4*, *amh*), C2 (*amh*, *dmrt1*, *wt1a*, *foxl2l*, *id1*, *smad9*, *zfp36l2*) or C3 (*gdf6Y*) probes, respectively. All washing steps during hybridization and development were carried out thrice for 5 min. The counter-staining with DAPI was either performed according to the manufacturer's protocol or by applying and mounting the slides with ProLong Gold antifade reagent with DAPI (Thermo Fisher Scientific). Whole tissue scans were acquired with a Plan-Apochromat 20×/0.8

M27 objective at the AxioScan.Z1 (Zeiss). Z-stacks were recorded with a Plan-Apochromat 40×/1.3 or 63×/1.4 Oil DIC M27 objective at the Axio Imager.Z2 with ApoTome.2 (Zeiss). Images were processed in ZEN 2.6 (Blue Edition, Zeiss) by combining all sections of a Z-stack into a maximum-intensity projection.

## Image analysis and minimal distance determination

The 3D image analysis pipeline was established using Arivis Vision4D (v4.1.2; now Zeiss Arivis). In z-stack images of 0 dph males, gonads were selected as regions of interest, exported as.czi-file, and used in the subsequent analysis. In Arivis Vision4D, several object classes were detected by the following segmentation modules: 1. Nuclei in the DAPI channel with Cellpose-based Segmenter. 2. The RNAscope signals in the three other fluorescence channels with blob detection (size parameter 0.147–23.4 µm). This resulted in 4 groups of objects (nuclei, *ddx4*, *amh*, *gdf6Y*). The detected objects with green fluorescence contained, besides *ddx4* positive ones, auto-fluorescent objects, which were excluded by size and shape. The minimal distance between each object and the respective closest object of each of the 4 groups was determined in 3D. The minimal distance of the *gdf6Y* to *ddx4* or *amh* objects served as a readout of proximity between the respective mRNAs. The shorter the minimal distance, the likelier the co-localization within the same cell.

## Cell culture for RNA-seq analyses and validation

TM4 (ECACC 88111401), HeLa (DSMZ ACC 57), and HEK293 (DSMZ ACC 305) cells were cultivated in Dulbecco's Modified Eagle Medium (DMEM; TM4, HeLa: Gibco™ #41966 – high glucose, pyruvate; HEK293: Gibco™ #31966 – high glucose, GlutaMAX™ Supplement, pyruvate) with 10% FBS at 37 °C and 5% $CO_2$. For transfections, $0.5 \times 10^6$ HeLa and TM4 cells or $4 \times 10^5$ HEK293 cells per well were seeded into 6- or 12-well plates, respectively. 24 h later, the cells were transfected with 2.5 µg or 1 µg of plasmid DNA (pcDgdf6Y, pcDgdf6Ydel9, or pcDNA3.1-HA) per well of a 6- or 12-well plate, respectively, using Lipofectamine 3000 (Thermo Fisher Scientific) and incubated for 24 h at 37 °C and 5% $CO_2$ before RNA isolation.

## RNA isolation

The QIAGEN RNeasy Mini Kit was used to isolate RNA from *N. furzeri* gonads and transfected cells. Killifish tissue was lysed in RLT buffer containing β-mercaptoethanol using the QIAGEN TissueLyzer II according to the manufacturer's instructions. Cells were lysed directly in the wells using RLT buffer containing ß-mercaptoethanol. Cell samples were further homogenized using the QIAshredder Kit (QIAGEN). DNase digestion was performed on-column using the RNase-Free DNase Set (QIAGEN) as suggested in the RNeasy Mini Kit protocol. The RNA was eluted in 50 µl DEPC-treated $H_2O$.

Total RNA from dechorionated embryos and hatchling trunks was extracted by adding TRIzol reagent (Thermo Fisher Scientific) and homogenization with the QIAGEN TissueLyser II. For phase separation, the lysate was incubated for 5 min at room temperature before adding 0.2 volumes of chloroform relative to the starting amount of TRIzol reagent. The solution was shaken vigorously for at least 15 s, incubated for 3 min at room temperature, and centrifuged for 20 min at $12,000 \times g$ at 4 °C. The aqueous phase was transferred into a fresh Tube and 1 volume of chloroform was added. Shaking, incubation at room temperature, centrifugation, and transfer of the aqueous phase into a fresh Tube were repeated. Alternatively, the lysate was transferred into Phasemaker tubes (Thermo Fisher Scientific), and the phase separation was performed in a single step according to the manufacturer's instructions. 0.5 µl GlycoBlue Coprecipitant (Thermo Scientific Scientific), 1.1 volumes of isopropanol, and 0.16 volumes of 2 M sodium acetate (pH 4.0) were added for RNA precipitation. After mixing and incubation for 10 min at room temperature, samples were centrifuged for 20 min at $12000 \times g$ and 4 °C. The supernatant was

removed and 500 µl of 80% ethanol were added to wash the pellet by vortexing and centrifugation at $7500 \times g$ for 10 min at 4 °C. The washing was repeated. The pellet was air-dried for 10–15 min and dissolved in 15–30 µl of DEPC-treated $H_2O$. For optimal dissolving of the pellet, the sample was incubated at 65 °C for 5 min. For RT-qPCR of *foxl2l*, samples were cleaned up and DNase-digested with the QIAGEN RNeasy Mini Kit and the RNase-Free DNase Set (QIAGEN) according to the manufacturer's instructions.

## Reverse transcription-quantitative PCR (RT-qPCR)

For cDNA Synthesis up to 1 µg of total RNA was used. The reverse transcription was performed using the iScript cDNA Synthesis Kit (Bio-Rad) according to the protocol provided by the manufacturer. RT-qPCR was performed in 384 well plates using the CFX384 Real-Time system and CFX Manager 3.1 (Bio-Rad). Each 10 µl reaction contained 5 µl SYBR GreenER™ qPCR SuperMix Universal (Thermo Fisher Scientific), forward and reverse primer at a concentration of 200 nM each, and up to 3 ng/µl cDNA and was cycled according to the manufacturer's instruction. Annealing temperatures were 60 °C or 63 °C for *N. furzeri* or human cell assays, respectively. Information on the analyzed *N. furzeri* genes is supplied in Supplementary Data 1g. Oligonucleotides and their applications are listed in Supplementary Table 6.

RT-qPCR results were evaluated using the determined threshold cycles (*Ct*) and primer efficiencies (*E*) by calculating either the relative expression (*R*) of a gene of interest (goi) to a reference gene (ref) with Eq. (1) in Supplementary Figs. 4a and 5d or the fold change (*FC*) of a goi based on the median *Ct* of a control group (*gdf6Y*: male, other goi: female) using one or two reference genes with Eq. (2) from Hellemanns et al. [76] (all other RT-qPCR data).

$$R = \frac{E_{ref}{}^{Ct,ref}}{E_{goi}{}^{Ct,goi}} \tag{1}$$

$$FC = \frac{E_{goi}{}^{\Delta Ct,goi}}{\sqrt[f]{\prod_0^f E_{ref_0}{}^{\Delta Ct,ref_0}}} \tag{2}$$

## RNA sequencing

Sequencing of RNA samples was performed using Illumina's next-generation sequencing methodology[77]. In detail, DNase-digested total RNA from TM4, and HeLa cells were quantified, and quality was checked using the Tapestation 4200 instrument in combination with an RNA ScreenTape (both Agilent Technologies). Libraries were prepared from 300 ng total RNA using NEBNext Ultra II Directional RNA Library Preparation Kit in combination with NEBNext Poly(A) mRNA Magnetic Isolation Module and NEBNext Multiplex Oligos for Illumina (Unique Dual Index UMI Adaptors RNA) following the manufacturer's instructions (New England Biolabs). Deviating from the instructions, qPCR was performed for each library before final amplification to determine the optimal number of cycles (exponential phase) to avoid over-amplification of the libraries in the final amplification step. The quantification and quality check of libraries was done using the Bioanalyzer 2100 instrument and the D5000 kit (Agilent Technologies). Libraries were pooled and sequenced on a NovaSeq6000. The system ran in 101 cycle/single-end/standard loading workflow mode (S1 100 cycle kit v1.5). Sequence information was converted to FASTQ format using bcl2fastq v2.20.0.422.

## RNA-seq data analyses

The published RNA-Seq data[8] (ENA: ERR879038 - ERR879056) from male and female *N. furzeri* at different developmental stages was filtered with sga [https://github.com/jts/sga] (parameters: preprocess -q

30 -m 50 –dust). Quality-passed reads were mapped to the *N. furzeri* reference genome[8] using STAR[78] (parameters: --alignIntronMax 100000). Counts per sample and gene were obtained with the summarizeOverlaps function of the R package GenomicAlignments[79]. DEGs (adjusted *p*-value < 0.05; Supplementary Data 1) between females and males of each developmental stage were derived using DESeq2[80]. DEGs were visualized in volcano plots generated with VolcaNoseR[81] [https://huygens.science.uva.nl/VolcaNoseR2/]. For the graphical representation of expression values, regularized logarithmic transformation was applied to the counts using the rlog function of DESeq2.

For the analysis of RNA-Seq data of murine TM4 and human HeLa cells (Supplementary Data 2, 3), the names of the RNA-Seq reads were extended by their unique molecular identifier (UMI) with UMI-tools' extract command version 1.1.1[82]. These modified reads were aligned with STAR 2.7.10a[78] (parameters: --alignIntronMax 100000, --outSJfilterReads Unique, --outSAMmultNmax 1, --outFilterMismatchNoverLmax 0.04) to the *M. musculus* (GRCm39) and *H. sapiens* (GRCh38) reference genomes, respectively, with the Ensembl genome annotation release 107. PCR duplicates were removed from the alignment results using UMI-tools's dedup command. For each gene, reads that map uniquely to one genomic position were counted with FeatureCounts 2.0.3 (multi-mapping or multi-overlapping reads were discarded, and stranded mode was set to –s 2)[83]. The pairwise comparisons of *gdf6Y*-expressing samples with either *gdf6Y^{del9}*-expressing or empty vector samples were analyzed for differential gene expression. Therefore, DEGs were determined with R 4.1.3 using the package DESeq2 1.34.0[80]. Only genes with at least one read count in any of the analyzed samples of a particular comparison were subjected to DESeq2. For each gene, the *p*-value was calculated using the Wald significance test. The resulting *p*-values were adjusted for multiple testing with the Benjamini & Hochberg correction (FDR). The log2 fold change (log2FC) values were shrunk with the DESeq2 function lfcShrink(type = "apeglm") to control for the variance of log2FC estimates for genes with low read counts[84]. Genes with an adjusted *p*-value < 0.05 are considered DEGs. Selected DEGs were visualized as Z-scores from DESeq2-normalized counts in a heatmap using Morpheus [https://software.broadinstitute.org/morpheus]. The overlap analysis of the up- or downregulated DEGs identified in TM4 cells with those identified in the same way in human HeLa cells was based on gene symbols and followed by an ortholog verification in Ensembl release 109.

Overrepresentation analyses of KEGG pathways were performed using WebGestalt 2019[85] [https://www.webgestalt.org/] (WEB-based GEne SeT AnaLysis Toolkit). For the *N. furzeri* RNA-Seq data, gene symbols from *D. rerio* orthologs assigned to the genebuild_v1.150922 *N.furzeri* annotation[8,86] (Supplementary Data 1) were used to search for enriched KEGG pathways in a *D. rerio* organismal background. Therefore, female, or male DEGs at the different developmental stages were compared to a reference list of all genes identified in the whole data set (Supplementary Data 1h). For the TM4 RNA-Seq data, gene symbols of the 42 selected DEGs were compared against the *M. musculus* genome to identify enriched KEGG pathways in that genetic background. Parameters for the enrichment analyses were a minimum number of 5 IDs in a category, a maximum number of 2000 IDs in a category, and the Benjamini & Hochberg correction for multiple testing.

### Smad-activation luciferase reporter assay

Medaka fibroblast (OLF-136, RIKEN BRC RCB0184) cells were cultured at 28 °C and 5% CO$_2$ in DMEM supplemented with high glucose (4.5 g/L), 4 mM glutamine, and 15% FBS[87]. For transfection, cells were grown to 80% confluency in six-well plates and subsequently transfected using the Amaxa™ Nucleofector™ II Device (Amaxa Biosystems, Cologne, Germany) with the Nucleofector® Kit C and the Program Y-001. In detail, cells were cultured until they reached approximately

60–80% confluency and harvested with trypsin-EDTA solution (59418C, Sigma-Aldrich). $2 \times 10^6$ cells were pelleted by centrifugation at $1000 \times g$ for 5 min and resuspended in 100 µL of Nucleofector® Solution with 2 µg of plasmid DNA of interest, according to the manufacturer's manual. After mixing, the suspension was transferred to a nucleofection cuvette, and Program Y-001 was applied. After adding 500 µl of culture medium in the cuvette, cells were immediately seeded in 6 well plates pre-filled with a medium at a density of 500,000 cells per well.

The luciferase reporter assay was performed as reported previously[23]. Medaka fibroblast cells were co-transfected with 400 ng of one out of four *gdf6*-expression plasmids (pcDgdf6X, pcDgdf6Y, pcDgdf6Ydel9, or pcDgdf6Yins25), 300 ng of one out of five expression plasmids for fusion proteins consisting of Smad activation and Gal4 DNA binding domains (Smad1-AD-GAL4-DBD, Smad2-ADGAL4-DBD, Smad3-AD-GAL4-DBD, Smad5-AD-GAL4-DBD, or Smad8-AD-GAL4-DBD), 300 ng of a firefly luciferase reporter plasmid under the control of a UAS sequence containing minimal promoter (UAS-luc), and 5 ng of a *Gaussia* luciferase expression plasmid for normalization (pCMV-Gluc) per well. After 24 h, cells were washed twice with phosphate-buffered saline, lysed with 75 µL of passive lysis buffer (Promega), and then subjected to the luciferase assay. Firefly luciferase activity was quantified using the Dual-Luciferase Reporter Assay System (Promega) and normalized against *Gaussia* luciferase activity.

### Statistics and reproducibility

Calculations, graphical presentation, and statistical analyses of non-sequencing data were realized using Microsoft Excel for Microsoft 365 MSO (Version 16.0.1.) 64-bit and GraphPad Prism version 9.0.2 for Windows, GraphPad Software, San Diego, California USA, www.graphpad.com. RT-qPCR data were Log2-transformed to allow the assumption of normal distribution within groups and tested using Welch's *t*-test or Welch's ANOVA followed by Dunnett's T3 multiple (two-tailed) comparisons tests to account for the differences in variance between the groups. These tests were also applied to data with an expected normal distribution e.g., weight and length measurements. Non-parametric tests were used where normal distribution could be excluded, e.g. when groups included virtual null or strongly inhomogeneous values. Test details are provided in the Source Data. Representative experiments were independently repeated twice with similar results.

### Reporting summary

Further information on research design is available in the Nature Portfolio Reporting Summary linked to this article.

### Data availability

The RNA-Seq data in this study have been deposited in NCBI's Gene Expression Omnibus[88] under accession codes GSE263626 (female and male *N. furzeri* samples at different life stages) and GSE233997 (human HeLa and murine TM4 cells). RNA-Seq data analysis results generated in this study are provided as Supplementary Data 1, 2, and 3 for *N. furzeri* samples, TM4 cells, and HeLa cells, respectively. The bisulfite sequencing data in this study have been deposited in NCBI's Sequence Read Archive[89] under accession code PRJNA974954. Source data are provided with this paper.

### Code availability

R Scripts used for RNA-Seq data analyses in this study have been deposited on GitHub and archived via Zenodo under accession code 13790201[90]. Python Scripts used for bisulfite sequencing data analyses in this study have been deposited on GitHub and archived via Zenodo under accession code 14186262[91].

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

## Acknowledgements

We thank Christina Ebert, Sabine Matz, Sabine Landmann-Weinsheimer, Linda Spilling, Silke Foerste, Peter Singer, Joseph Kölbel, and Gabriele Günther for their technical support, and Hanna Reuter for her bioinformatics expertise. We furthermore thank André Scherag for his statistical advice and Tehila Atlan, Itamar Harel, and Manfred Schartl for their helpful discussions. We are grateful to members of the Core Facilities Next Generation Sequencing (Ivonne Goerlich, Cornelia Luge, and Martin Bens), Life Science Computing, and Imaging, particularly Birgit Perner, for their contributions. We thank the members of FLI's fish facility, notably Simone Gruner, Martin Neumann, Johannes Wilfert, Marcus Schmidt, Clemens Peters, Benjamin Otto, Uta Naumann, and Beate Hoppe. This study was supported by the German Research Foundation (DFG) with the project number 428793369 to C.E. The FLI is a member of the Leibniz Association and is financially supported by the Federal Government of Germany and the State of Thuringia. The publication of this article was funded by the Open Access Fund of the Leibniz Association and the Leibniz Institute on Aging – Fritz Lipmann Institute (FLI), Jena, Germany. Funding Open Access funding enabled and organized by Projekt DEAL.

## Author contributions

Conceptualization: A.R., C.E.; software: R.S., P.K., A.P.; Formal Analysis: A.R., R.G., P.K., A.P., T.K.; investigation: A.R., H.M., M.T., M.K., R.G., C.A., N.H., A.H.; data curation: A.R., H.M., M.T., M.K., R.G., C.A., P.K., M.G., A.P.; writing – original draft: A.R.; writing – review & editing: A.R., C.E., H.M., P.K., R.S, A.P., A.H., M.K., C.A., M.G., N.H., R.G., T.K.; visualization: A.R.; supervision: C.E.; funding acquisition: C.E.

## Funding

## Competing interests

The authors declare no competing interests.
