## [Peer Review file · Nature Communications]

The master male sex determinant Gdf6Y of the turquoise killifish arose through allelic neofunctionalization

Corresponding Author: Dr Annekatrin Richter

Version 0:

Reviewer comments:

Reviewer #1

(Remarks to the Author)

In this manuscript, Richter et al uncover the Y-chromosome linked copy of the African turquoise killifish *Nothobranchius furzeri* Gdf6y as the male determining gene in this species. Indeed, sex chromosome pair evolution started relatively recently in this species, and the genetic driver of male differentiation had not been previously identified. Here, the authors show that deletion of Gdf6y (with mutant Y chromosome coded as Y* hereafter) leads to complete male-to-female sex-reversal as early as embryogenesis: XY* (phenofemale) fish develop female coloring patterns, functional ovaries with female-like expression of sex markers (e.g. Foxl2, Cyp19a1a) and are fertile. However, breeding of XY and XY* animals led to occurrence of malformations and improper embryonic development - especially YY* animals, potentially due to missing an intact copy of Gdf6x. Interestingly, Gdf6y is expressed in a small subset of Sertoli-like cells, and repressed induction of the master regulator of ovarian fate Foxl2 in the killifish embryonic ovary. Altogether, the authors provide interesting mechanistic evidence of the new role of a neofunctionalized copy of Gdf6 on the killifish Y chromosome as both necessary and sufficient for male sex-determination in this species.

Overall, the overarching conclusion of the manuscript on the role of Gdf6y is generally supported by the presented data, and will be of broad interest to both the sex-determination field and the growing turquoise killifish research community. However, some important points need to be addressed for clarity, rigor and reproducibility, with missing methodological information or inaccurate use of statistics as described below.

Major Comments:

1. Fig S5 The authors state that DEGs were defined as those genes with $\text{padj} < 0.05$ and do not mention thresholding on \log_2 fold change, although the volcano plots in Figure S5 suggest a threshold on absolute $\log_2\text{FC} > 1$. Is that what was done also in Figure 5A? Fold-change filtering, if not implemented into the statistical model used to identify DEGs, leads to poor FDR control and is not appropriate (see PMC2654802). Since fold change filtering leads to statistical issues, all of its use should be removed. In addition, the authors should make sure the methods appropriately describe the analyses presented. Finally, all R scripts used to perform omic analyses should be provided on github for long term reproducibility of the analyses (this reviewer could not find a reference to code deposition in the current manuscript).
2. Use of FPKM extracted values and graphpad prism on selected handpicked genes for differential gene expression (see line 642-643 in methods; Fig 4A-B) is completely unacceptable statistically. FPKMs are not appropriate measures to compare expression between samples (see for example <https://translational-medicine.biomedcentral.com/articles/10.1186/s12967-021-02936-w>). The authors should report VST normalized \log_2 count values from DESeq2 and the DESeq2 Wald test FDR for differential gene expression. If this was not significant, these results need to be removed from the paper, as FPKM/selected gene testing in prism is not statistically sound.
3. Some experiments have rather variable results but very small sample size (e.g. Fig 4E). Thus, due to sampling bias at that sample size, some results may be spurious (despite significant "p"). We recommend that enough samples be included (usually 5 or more per group), or conclusions should be significantly toned down for accuracy. In addition, many results are shown without information about replication or quantification (Figure 3 – how many independent animals was this staining done on? Was this quantified?) – for reproducibility, all data points, including from imaging, should be explicitly annotated with the number of independent animals assessed and quantification should be provided.
4. The paper makes extensive use of parametric tests (e.g. ANOVA), when the data looks clearly non normally distributed (e.g. Fig 1E-F, Fig 2A, Fig 4A,B,E, Fig 5D), which is a prerequisite for using parametric tests. We recommend that all statistics be redone using non parametric tests for robustness and accuracy.

5. Line 459: the authors mention they used lipofectamine or FuGENE for transfections, but don't explain which experiment used which one, how much DNA was used per transfection, how many cells were transfected, etc. In general, the methods should be revised to be more complete and more detailed for long-term reproducibility and accuracy.
6. The manuscript compares "adult" fish to 0 dph in the whole paper. However, the authors never define at what age a fish becomes an adult, nor describe at what stage of adulthood the fish was at when they perform experiments (at the exception of Figure 5, describing the use of 3-month-old animals). This information needs to be explicitly included for the paper (if the same age was used for all experiments) or for each experiment (if ages varied among experiments).

Minor Comments:

1. The authors should indicate how many eggs were injected to produce the results they report in terms of transgenesis.
2. The authors should indicate how many offspring the XY* -XY pairings produced (from how many pairs tested, at what age) and how many of the fertilized eggs were viable.
3. They did RNAscope on *gdf6y* on XX and had no signal. I think doing RNAscope of *gdf6y* on XY* would have supported more of their findings (Maybe they did and didn't include it).
4. Line 428/429: the authors state they used the GRZ strain for most experiment, but that they change the strain for the in-situ hybridization to MZM0403. Although the authors are transparent about the change, it would be important to explain why the in situ was not performed on the GRZ strain.

Reviewer #2

(Remarks to the Author)

The manuscript, by Richter et al, tried to prove that a Y-allelic *gsdf6* (*gsdf6Y*) is the sex determination gene in turquoise fish, and analyzed the candidate genes that are possibly regulated by *gsdf6Y*. To demonstrate that *gsdf6Y* is the sex determination gene of turquoise fish, authors tried inducing the mutations specific to *gsdf6* on the Y chromosome and found that a certain ratio of mosaic XY* fish increases female-specific genes with female appearance. This suggests that *gsdf6Y* is the sex determination gene of turquoise fish. In addition, authors discuss the evolutionary origin of *gsdf6Y* and conclude that it was created as neofunctionalization. Since it is described in abstract, it would be a main context of this manuscript and the reviewer agrees with this notion. However, allelic diversification has been often reported for many sex determination genes such as pufferfish, dancena medaka and pejerrey. In addition, there are several points to be addressed.

1. The reviewer wonders why the authors did not analyze mutants in F1 generation, which is very straightforward and gives clear results. Was that because the mutations in a Y allele displayed embryonic lethal? In mosaic fish, it cannot be ruled out that expression of *gsdf6Y*-responsive genes is a side-effect by genome editing although authors argue that the mutations are highly specific (but to alleles on Y and X).
2. Regarding expression, authors should show the initial timing of *gsdf6Y* expression by in situ hybridization or RT-PCR using RNA extracted from gonads.
3. Authors state that several genes are identified as *gsdf6Y*-responsive genes, which are likely located downstream of *gsdf6Y* function. These genes include components of TGF β pathway, *foxl2l* and *zpf36l2*. But again, identification is not straightforward. In addition, there is no wonder that the components of TGF β pathway are listed because another TGF β ligand and receptor genes have been known to function in sex determination pathways, such as *amhY* and *gsdfY*. The reviewer would say that in this manuscript there is no new functional insight on what occurs in the pathway of *gsdf6Y*.
4. Regarding *zpf36l2*, authors should provide more solid evidences showing that *zpf36l2* is involved in the sex determination.
5. Is the possible receptor for GSDF6Y expressed in somatic cells or germ cells? This is going to be one of the important and novel points promoting understanding of the mechanism of GSDF6Y-mediated sex determination. This would make the manuscript more valuable for those in this field.

Version 1:

Reviewer comments:

Reviewer #1

(Remarks to the Author)

The authors have addressed most of my previous comments satisfactorily. I believe conclusions are now sufficiently supported by the presented evidence

Reviewer #2

(Remarks to the Author)

The authors provide point to point replies to my concerns.

Especially to address roles of *gsf6Y*, authors created and analyzed new CRISprants of downstream genes (*id1* and *zfp3612*). Although the images of FigS7 are not clear enough to see *foxl2l* expression in germ cells, authors found that expressions of some male genes are downregulated and partial sex reversal was recognized in the CRISprants. This suggests that *gsf6Y* is epistatically located upstream of these genes and reveals a mechanistic role of *gsdf6Y*, which is appreciated in the revised manuscript.

Therefore, I agree with their contention that *gsdf6Y* is a master sex determination gene in a turquoide fish.

In a view point of development and cellular signaling exerted by *gsdf6Y*, however, there are previous examples that *id* genes are under regulation by TGF β ligands. In addition, the involvement of *id* (*id2bb*) gene in sex determination is reported in another teleost fish, pirarucu (Adolfo et al., Scientific Reports 2021), which supports a general (and conventional) idea that the TGF signaling pathway could be one of the major stems (masculinization) of sex determination mechanism.

Regarding appearance of a new actor, *gsdf6Y*, this manuscript contributes to adding a new repertoire to a variety of sex determination genes. But the new actor is not surprising because there are many TGF related genes reported to act as a sex determination gene. Therefore a reviewer would say there is no wonder a new *gsdf* has been picked up evolutionally and established as a sex determination gene in some species.

Evolutional emergence of sex determination gene by allelic diversification is also reported in many vertebrate species as a reviewer described in the original comment.

The reviewer would regrettably say that the revised manuscript and the replies do not go beyond such general points mentioned above.

We thank the reviewers for their constructive comments (*in italics*) and have addressed the issues that were raised in the following manner.

Reviewer #1 (Remarks to the Author):

*In this manuscript, Richter et al uncover the Y-chromosome linked copy of the African turquoise killifish *Nothobranchius furzeri* *Gdf6y* as the male determining gene in this species. Indeed, sex chromosome pair evolution started relatively recently in this species, and the genetic driver of male differentiation had not been previously identified. Here, the authors show that deletion of *Gdf6y* (with mutant Y chromosome coded as *Y** hereafter) leads to complete male-to-female sex-reversal as early as embryogenesis: *XY** (phenofemale) fish develop female coloring patterns, functional ovaries with female-like expression of sex markers (e.g. *Foxl2*, *Cyp19a1a*) and are fertile. However, breeding of *XY* and *XY** animals led to occurrence of malformations and improper embryonic development - especially *YY** animals, potentially due to missing an intact copy of *Gdf6x*. Interestingly, *Gdf6y* is expressed in a small subset of Sertoli-like cells, and repressed induction of the master regulator of ovarian fate *Foxl2* in the killifish embryonic ovary. Altogether, the authors provide interesting mechanistic evidence of the new role of a neofunctionalized copy of *Gdf6* on the killifish Y chromosome as both necessary and sufficient for male sex-determination in this species.*

*Overall, the overarching conclusion of the manuscript on the role of *Gdf6y* is generally supported by the presented data, and will be of broad interest to both the sex-determination field and the growing turquoise killifish research community. However, some important points need to be addressed for clarity, rigor and reproducibility, with missing methodological information or inaccurate use of statistics as described below.*

We very much thank the reviewer for appreciating our work. To resolve a minor misunderstanding we have restructured and rephrased parts of the manuscript. To clarify that the *YY** phenotype is caused by the Y chromosomal inactivation instead of the lack of *gdf6X*, we rephrased the chapter “*YY** embryos develop anomalies and die until the hatching stage” (specifically lines 142-160) and moved the *gdf6X*-inactivation to the new, subsequent chapter “*Gdf6Y* compensates for the loss of *gdf6X*” allowing the accommodation of new data regarding *gdf6X*-deficient animals (Fig. 2 e-j, Fig. S3 h-j, and lines 184-196).

Major Comments:

1. *Fig S5*The authors state that DEGs were defined as those genes with $padj < 0.05$ and do not mention thresholding on \log_2 fold change, although the volcano plots in Figure S5 suggest a threshold on absolute $\log_{FC} > 1$. Is that what was done also in Figure 5A? Fold-change filtering, if not implemented into the statistical model used to identify DEGs, leads to poor FDR control and is not appropriate (see PMC2654802). Since fold change filtering leads to statistical issues, all of its use should be removed. In addition, the authors should make sure the methods appropriately describe the analyses presented. Finally, all R scripts used to perform omic analyses should be provided on github for long term reproducibility of the analyses (this reviewer could not find a reference to code deposition in the current manuscript).

All fold change filtering was removed from the presented transcriptomics data. This concerns the data presented as volcano plots in Fig. 4 a (formerly presented in Fig. S5 A) and as a heatmap in Fig. 6 a (formerly presented in Fig. 5 A). The definition of DEGs with an adjusted p-value of $FDR < 0.05$ is stated in the respective figure legends and the METHODS section, lines 766-768 and 794-795. The R scripts applied for the data analyses were provided on the citable platform Zenodo including a link to the respective GitHub deposition. Please find the references in the revised DATA AND CODE AVAILABILITY section, lines 842-852.

2. *Use of FPKM extracted values and graphpad prism on selected handpicked genes for differential gene expression (see line 642-643 in methods; Fig 4A-B) is completely unacceptable statistically. FPKMs are not appropriate measures to compare expression between samples (see for example <https://translational-medicine.biomedcentral.com/articles/10.1186/s12967-021-02936-w>). The authors should report VST normalized log2 count values from DESeq2 and the DESeq2 Wald test FDR for differential gene expression. If this was not significant, these results need to be removed from the paper, as FPKM/selected gene testing in prism is not statistically sound.*

We removed all plots depicting RPKM with the statistical tests in question and now present rlog-transformed log2 count values as a VST-transformation equivalent with the DESeq2 Wald test FDR instead (Fig. S4 h, S7 a, S7 b, and S8 d formerly Fig. S3 H, S6 A, 4 A, and S7 D respectively). While this exchange slightly changed the result of the analyses (lines 385-393), our conclusions stayed overall consistent.

3. *Some experiments have rather variable results but very small sample size (e.g. Fig 4E). Thus, due to sampling bias at that sample size, some results may be spurious (despite significant "p"). We recommend that enough samples be included (usually 5 or more per group), or conclusions should be significantly toned down for accuracy. In addition, many results are shown without information about replication or quantification (Figure 3 – how many independent animals was this staining done on? Was this quantified?) – for reproducibility, all data points, including from imaging, should be explicitly annotated with the number of independent animals assessed and quantification should be provided.*

We understand the reviewer's concern regarding the sample sizes and adjusted where feasible. However, in experiments including embryonic samples (0 dph or younger) e. g., Fig. 4 e, it is not possible to predetermine group size (defined by sex and genotype) due to blind sampling. Hence, we aimed to inject as many zygotes as available for such experiments (lines 631-632, Supplementary Tab. S3), performed two repetitions that gained similar results, and present data with group sizes of at least 3. Furthermore, we included repetitions of the experiments shown in Fig. 1 f-g (formerly Fig. 1 E-F, $n \geq 3$) with animals from subsequent generations (Fig. S2 e-f, $n=5$), and explicitly state the number of samples for all experiments in the figures or legends. Regarding RNAscope experiments, the complex cellular structure of the gonadal tissue obstructed an automated assignment

of signals to certain cells and thereby a meaningful quantification in the sense of our research question, i.e., in which cells the analyzed genes are expressed. Instead, we performed a proximity analysis in 3D on one batch of embryonic samples (0 dph, n=4) to support our main finding of *gdf6Y* expression in pre-Sertoli cells (Fig. 3 c, lines 218-221 and 677-688).

4. *The paper makes extensive use of parametric tests (e.g. ANOVA), when the data looks clearly non normally distributed (e.g. Fig 1E-F, Fig 2A, Fig 4A,B,E, Fig 5D), which is a prerequisite for using parametric tests. We recommend that all statistics be redone using non parametric tests for robustness and accuracy.*

We agree with the reviewer that the use of parametric tests does not apply to some data sets e.g., Fig. 2 I (formerly Fig. 2 F) and the new Fig. 3 c, where we now use non-parametric tests. However, we would argue that log-normal distribution can be assumed within individual groups of RT-qPCR data, as those represent gene expression on RNA level, which is usually assumed as log-normally distributed, e. g. in case of RNA-Seq evaluation by DESeq2¹. Therefore, all our RT-qPCR data is displayed and tested as \log_2 (fold change). Nevertheless, we understand that the variances within the individual experimental groups in those data sets differ widely (illustrated by indicating the mean and standard deviation in the respective graphs), particularly, when comparing male and female gonads (Fig. 1 f-g, formerly Fig. 1 E-F). Consequently, conventional *t*-tests and ANOVA are indeed not well suited to compare these experimental groups, as they assume equal variance in addition to normal distribution. Hence, we consulted the advice of a statistician (André Scherag; Institute of Medical Statistics, Computer and Data Sciences; Jena University Hospital/Friedrich-Schiller-University Jena; Jena; Germany) to improve our statistical processing. He suggested the use of tests that assume normal distribution but unequal variance like Welch's *t*-test and Welch's ANOVA, which are now applied to almost all experimental RT-qPCR data sets (e. g., Fig. 1 f-g, S7 c, 4 e, and 6 d, formerly Fig. 1 E-F, 4 B, 4 E, and 5 D respectively). The exception is the newly added *zfp36l2* CRISPR assay data set (Fig. 6 f and S9 c). This experiment like all CRISPR assays (explained in Fig. 4 d) involves highly variable experimental groups namely mosaic XX, XY*, or XY animals for the respective inactivated gene (here: *zfp36l2*). In other CRISPR assays this led to a high variance within the mosaic groups for most of the analyzed genes and we adhered to Welch's ANOVA. In the case of the *zfp36l2* CRISPR assay, however, the experimental group in question (XX *zfp36l2* CRISPRs) showed a bimodal distribution for 5 out of 6 analyzed genes and we switched to the Kruskal-Wallis test as a non-parametric alternative. Our statistical considerations are also briefly summarized in lines 833-839.

5. *Line 459: the authors mention they used lipofectamine or FuGENE for transfections, but don't explain which experiment used which one, how much DNA was used per transfection, how many cells were transfected, etc. In general, the methods should be revised to be more complete and more detailed for long-term reproducibility and accuracy.*

In response to the reviewer's comment, we revised all method descriptions and restructured the METHODS and SUPPLEMENTARY METHODS sections. To address the specific issue raised in the reviewer's comment, we moved the chapter concerning cell culture procedures before the chapters describing RNA-based methods and renamed it "Cell culture for RNA-Seq analyses and validation" (lines 691-698). This now includes the necessary information regarding the cell culture and transfections for the cell line-based RNA-Seq and its validation in HEK293 cells. The information regarding medaka fibroblast cell culture and transfection was moved to the paragraph "Smad-activation luciferase reporter assay" (lines 809-830) and the transfection method was corrected to nucleofection using the Amaxa™ Nucleofector™ II Device (Amaxa Biosystems, Cologne, Germany) with the Nucleofector® Kit C and the Program Y-001 (details see lines 809-819).

6. *The manuscript compares "adult" fish to 0 dph in the whole paper. However, the authors never define at what age a fish becomes an adult, nor describe at what stage of adulthood the fish was at when they perform experiments (at the exception of Figure 5, describing the use of 3-month-old animals). This information needs to be explicitly included for the paper (if the same age was used for all experiments) or for each experiment (if ages varied among experiments).*

We addressed the issue raised by the reviewer by explicitly stating the age of all used animals in the respective figures or figure legends. We furthermore exchanged the term "adult" by "sexually mature" in most instances, as this is the relevant criterion for our work, and added additional timing information regarding those terms in lines 50-52 and line 585.

Minor Comments:

1. *The authors should indicate how many eggs were injected to produce the results they report in terms of transgenesis.*

The required information is now included in detail in Supplementary Tab. S3. A statement mentioning that we inject up to 324 zygotes per batch of collected eggs was also included in the METHODS section (lines 631-632). This number is defined by the time window of the 1-cell embryo stage and the capacity of the injection mold we use (described in Krug *et al.*²).

2. *The authors should indicate how many offspring the XY* -XY pairings produced (from how many pairs tested, at what age) and how many of the fertilized eggs were viable.*

To address this point, we included two additional experiments in the manuscript. One shows the genotype and numbers of phenofemale F1 offspring at one month of age and older (Fig. 1 d) and the second demonstrates the reproductive output of XY* phenofemales

from our propagated lines in comparison to their XX sisters (Fig. S2 g-h). This data is discussed in lines 116-118 and 127-129, respectively.

3. *They did RNAscope on gdf6y on XX and had no signal. I think doing RNAscope of gdf6y on XY* would have supported more of their findings (Maybe they did and didn't include it).*

Respective RNAscope data is now included in Fig. S4 b and mentioned in lines 214-217.

4. *Line 428/429: the authors state they used the GRZ strain for most experiment, but that they change the strain for the in-situ hybridization to MZM0403. Although the authors are transparent about the change, it would be important to explain why the in situ was not performed on the GRZ strain.*

The respective *in situ* hybridization data (Fig. S4 i, formerly S3 G) was generated before we decided on GRZ-D as our strain of choice because we had the most continuous Y chromosomal sequence information from GRZ-D due to the BAC library³. While MZM0403 has the smallest sex-determining region (SDR) on the Y chromosome³, the sex-determining mechanism is conserved in all *N. furzeri* strains³ and the sister species *N. kadleci*⁴, which we now also state in the METHODS section, lines 580-583. Hence, the choice of strain should not influence the result of the experiment.

Reviewer #2 (Remarks to the Author):

The manuscript, by Richter et al, tried to prove that a Y-allelic gsdf6 (gsdf6Y) is the sex determination gene in turquoise fish, and analyzed the candidate genes that are possibly regulated by gsdf6Y. To demonstrate that gsdf6Y is the sex determination gene of turquoise fish, authors tried inducing the mutations specific to gsdf6 on the Y chromosome and found that a certain ratio of mosaic XY fish increases female-specific genes with female appearance. This suggests that gsdf6Y is the sex determination gene of turquoise fish. In addition, authors discuss the evolutionary origin of gsdf6Y and conclude that it was created as neofunctionalization. Since it is described in abstract, it would be a main context of this manuscript and the reviewer agrees with this notion. However, allelic diversification has been often reported for many sex determination genes such as pufferfish, dancena medaka and pejerrey.*

Regarding this summary of our work an important clarification is required. Reviewer #2 assumes that *gdf6Y* is an ortholog of *gsdf* instead of *gdf6*. While both encode TGF- β ligands, they have very different gene histories. *Gsdf* is an important but fish-exclusive factor in gonadal development. *Gdf6* on the other hand is conserved in vertebrates and is a crucial developmental gene in axial, joint, and eye formation but without any known function in the reproductive system. Thus, there is a lot of novelty in the identification and characterization of *gdf6Y* as a male sex-determining factor in killifish.

In addition, there are several points to be addressed.

1. *The reviewer wonders why the authors did not analyze mutants in F1 generation, which is very straightforward and gives clear results. Was that because the mutations in a Y allele displayed embryonic lethal? In mosaic fish, it cannot be ruled out that expression of gsdf6Y-responsive genes is a side-effect by genome editing although authors argue that the mutations are highly specific (but to alleles on Y and X).*

We are grateful that the reviewer raised our awareness that this essential manuscript section is unclear. To address this issue, we explicitly state the respective generation where necessary and added new data of XY* phenofemales from several generations (Fig. 1 d and new Fig. S2, discussed in lines 116-129) to illustrate the robustness of the phenotype. To clarify the cause of the observed lethality of YY* embryos, which constitute one-quarter of the phenofemales' offspring due to Mendelian ratios, we restructured and rephrased the section concerning the YY* embryos. We moved the *gdf6X*-inactivation from the second chapter "YY* embryos develop anomalies and die until the hatching stage" to a new third chapter "*Gdf6Y* compensates for the loss of *gdf6X*" to clarify that the YY* phenotype is caused by Y chromosomal inactivation (rephrased lines 142-160) instead of the lack of *gdf6X* and to accommodate new data regarding *gdf6X*-deficient animals (Fig. 2 e-j, Fig. S3 h-j, and lines 184-196), respectively.

2. *Regarding expression, authors should show the initial timing of gsdf6Y expression by in situ hybridization or RT-PCR using RNA extracted from gonads.*

To address this comment, we performed both *in situ* hybridization on one and RT-qPCR on eight embryonic stages (Fig. S6 f-h, lines 269-282). Of note, both methods have some limitations. On the one hand, the conventional *in situ* hybridization probes we use for whole-mount analyses detect both *gdf6X* and *gdf6Y* mRNAs because of their high similarity. Hence, we have chosen to analyze a developmental stage that allows the inclusion of YY* embryos to exclusively detect *gdf6Y* transcripts in those samples. On the other hand, it is virtually impossible to isolate gonadal tissue from *N. furzeri* embryos using the currently available methodology, although we can specifically detect *gdf6Y* in RT-qPCR. While our data support the finding that *gdf6Y* in addition to sex determination fulfills the same developmental functions as *gdf6X*, the data do not pinpoint the initial timing of sex determination. However, analyzing *id1* as *gdf6Y*'s most upregulated downstream agent, we obtained data supporting that sex determination is initiated at the pharyngula stage at 10 days post-fertilization or earlier (Fig. 6 d, lines 390-393).

3. *Authors state that several genes are identified as gsdf6Y-responsive genes, which are likely located downstream of gsdf6Y function. These genes include components of TGFβ pathway, foxl2l and zpf36l2. But again, identification is not straightforward. In addition, there is no wonder that the components of TGFβ pathway are listed because another TGFβ ligand and receptor genes have been known to function in sex determination pathways, such as amhY and gsdfY. The reviewer would say that in this manuscript there is no new functional insight on what occurs in the pathway of gsdf6Y.*

The reviewer criticizes our approaches to identifying *gdf6Y*-responsive genes as not straightforward. However, we disagree with this notion and would argue that we employed two reasonable approaches. In our first approach, we analyzed available RNA-seq data to learn about the timing and processes that occur in *N. furzeri*'s early sexual development. We discovered that female germ cell proliferation, meiosis, and oogenesis are the first processes of sexual development that are detectable in embryonic samples. Among the identified genes related to these processes was *foxl2l*, which was previously shown to be an intrinsic determiner of the female germ cells in the medaka⁵ and zebrafish⁶. We localized its expression to early oocytes using RNAscope (Fig. S7 d, f) and confirmed its role in female germ cell sex determination downstream of *gdf6Y* in *N. furzeri* by analyzing oogenesis markers in mosaic *foxl2l* mutants (Fig. 4 e, S7 h-i). This served as proof of the principle that our CRISPant assay (Fig. 4 d) is suitable for the validation of candidate genes involved in sexual differentiation. In our second approach, we directly measured the transcriptional response to Gdf6Y signaling in two cell lines (Fig. 6 a-c), of which one is a model for the *gdf6Y*-expressing Sertoli cells (Fig. 3). The overlay of these results also contained genes involved in TGF- β signaling, particularly *id1*, *id2*, and *id3*, which are neither TGF- β ligands nor receptors but downstream effectors of TGF- β signaling in the targeted cells. This strongly suggests that activation of inhibitor of DNA-binding genes at the right place and time is necessary to initiate male sex determination in systems depending on TGF- β ligands and receptors. Hence, their identification is the next step in unraveling the mechanism behind those sex-determining genes. To this end, we performed additional RNAscope (Fig. S9 a) and a CRISPant assay (Fig. 6 e and S9 b) to verify the potential role of the most strongly upregulated inhibitor of DNA-binding gene *id1* (discussed in lines 398-430). In brief, the CRISPant assay hinted at increased meiosis in *id1* mosaic males but not to female levels, which was probably obstructed by a simultaneous, negative effect on female development as seen in XX CRISPants. This was consistent with the expression of *id1* in some female germ cells and somatic supporting cells of both sexes at the same developmental stage. Interestingly, male supporting cells are usually the place of initial sex determination in XX/XY systems. This suggests that *id1* is involved in male sex determination at an earlier stage, for which we found evidence by observing increased *id1* expression in males earlier in development (Fig. 6 d). This would place *id1* between *gdf6Y* and *foxl2l*, whose repression is likely a secondary effect mediated by determined Sertoli cells in males.

4. Regarding *zfp36l2*, authors should provide more solid evidences showing that *zfp36l2* is involved in the sex determination.

To this end, we performed RNAscope (Fig. S9 a) and a CRISPant assay (Fig. 6 f and S9 c) to verify the potential role of *zfp36l2* in sex determination (discussed in lines 398-430). Indeed, we observed a female-to-male sex reversal in some XX CRISPants and little *zfp36l2* expression in male gonadal somatic and Sertoli cells at the hatching stage and in adult testes.

5. *Is the possible receptor for GSDF6Y expressed in somatic cells or germ cells? This is going to be one of the important and novel points promoting understanding of the mechanism of GSDF6Y-mediated sex determination. This would make the manuscript more valuable for those in this field.*

In a new BAC transgenesis approach to validate *gdf6Y*'s sufficiency for sex determination, we observed a partial female-to-male sex reversal in mosaic *gdf6Y*-BAC transgenic XX animals (phenomales, Fig. 5 and lines 326-362). Regarding the question about Gdf6Y's target cells, F0 phenomales lost their germ cells probably due to an effect of mosaicism. We also identified structures reminiscent of testicular tubules with hypertrophic Sertoli cells in a phenomale's streak gonad. These data strongly suggest that Gdf6Y executes its sex-determining function by signaling to the male somatic supporting cells in an auto- or paracrine manner. This finding is further supported by recent publications reporting that germ cell removal in *N. furzeri* does not lead to a sex reversal in either direction^{7,8}. Hence, there is no male somatic predisposition in *N. furzeri* like in other small laboratory fish^{9,10}.

References

1. Love MI, Huber W, Anders S. Moderated estimation of fold change and dispersion for RNA-seq data with DESeq2. *Genome biology* **15**, 550 (2014).
2. Krug J, Perner B, Albertz C, Morl H, Hopfenmuller VL, Englert C. Generation of a transparent killifish line through multiplex CRISPR/Cas9mediated gene inactivation. *eLife* **12**, e81549 (2023).
3. Reichwald K, *et al.* Insights into Sex Chromosome Evolution and Aging from the Genome of a Short-Lived Fish. *Cell* **163**, 1527-1538 (2015).
4. Stundlova J, *et al.* Sex chromosome differentiation via changes in the Y chromosome repeat landscape in African annual killifishes *Nothobranchius furzeri* and *N. kadleci*. *Chromosome Res* **30**, 309-333 (2022).
5. Nishimura T, *et al.* Sex determination. *foxl3* is a germ cell-intrinsic factor involved in sperm-egg fate decision in medaka. *Science (New York, NY)* **349**, 328-331 (2015).
6. Liu Y, *et al.* Single-cell transcriptome reveals insights into the development and function of the zebrafish ovary. *eLife* **11**, e76014 (2022).
7. Abe K, *et al.* Sex-dependent regulation of vertebrate somatic growth and aging by germ cells. *Sci Adv* **10**, eadi1621 (2024).
8. Moses E, *et al.* The killifish germline regulates longevity and somatic repair in a sex-specific manner. *Nature aging* **4**, 791-813 (2024).
9. Kurokawa H, *et al.* Germ cells are essential for sexual dimorphism in the medaka gonad. *Proc Natl Acad Sci U S A* **104**, 16958-16963 (2007).

10. Slanchev K, Stebler J, de la Cueva-Mendez G, Raz E. Development without germ cells: the role of the germ line in zebrafish sex differentiation. *Proc Natl Acad Sci U S A* **102**, 4074-4079 (2005).

We thank the reviewer for her/his comments (*in italics*) and have addressed the points in this reply or the manuscript.

Reviewer #2 (Remarks to the Author):

The authors provide point to point replies to my concerns.

Especially to address roles of gsf6Y, authors created and analyzed new CRISprants of downstream genes (id1 and zfp36l2). Although the images of FigS7 are not clear enough to see foxl2l expression in germ cells, authors found that expressions of some male genes are downregulated and partial sex reversal was recognized in the CRISprants. This suggests that gsf6Y is epistatically located upstream of these genes and reveals a mechanistic role of gsdf6Y, which is appreciated in the revised manuscript.

Therefore, I agree with their contention that gsdf6Y is a master sex determination gene in a turquoide fish.

We agree that *foxl2l* expression by specifically female germ cells is hard to see in the images provided due to the germ cell marker (*ddx4*) signal intensity gradient within the pictures. This is caused by the strong expression of *ddx4* in the definitive oocytes. To convincingly demonstrate the expression of *foxl2l* by female germ cells before oogenesis, we included insets with zoom-in pictures of the *foxl2l*-positive germ cells with linearly increased contrast in the *ddx4* channel excluding the oversaturated signals of the definitive oocytes in Supplementary Fig. 7d and f.

In a view point of development and cellular signaling exerted by gsdf6Y, however, there are previous examples that id genes are under regulation fo TGFbeta ligands. In addition, the involvement of id (id2bb) gene in sex determination is reported in another teleost fish, pirarucu (Adolfo et al., Scientific Reports 2021), which supports a general (and conventional) idea that the the TGF signaling pathway could be one of the major stems (masculinization) of sex determination mechanism.

The study mentioned is referenced and discussed in the manuscript's discussion section.

Regarding appearance of a new actor, gsdf6Y, this manuscript contributes to adding a new repertoire to a variety of sex determination genes. But the new actor is not surprising because there are many TGF related genes reported to act as a sex determination gene. Therefore a reviewer would say there is no wonder a new gsdf has been picked up evolutionally and established as a sex determination gene in some species.

Evolutional emergence of sex determination gene by allelic diversification is also reported in many vertebrate species as a reviewer described in the original comment.

The reviewer would regrettably say that the revised manuscript and the replies do not go beyond such general points mentioned above.

While it is true that TGF- β signaling-related genes were repeatedly reported to act as sex-determining genes, there is little insight into how they intercept with the known mechanisms occurring during sex determination, particularly because such genes are expressed in both germ and various somatic cells during gonadal development¹. Although the transcription factors DMRT1 and FOXL2 play central roles in male and female sex determination, respectively, by acting in somatic supporting cells in all vertebrates², the gonadal germ cell content is crucial for the sexual fate decision in some teleost species, namely the widespread model organisms *Danio rerio*³ and *Oryzias latipes*⁴. To our knowledge, we show for the first time experimentally that a TGF- β signaling-related gene determines the male sex germ cell independently in the somatic supporting cells of a teleost species through means of known TGF- β signaling downstream effectors like the inhibitors of DNA binding and a new one, the post-transcriptional regulator Zfp36l2, likely globally changing gene expression. While it exceeds the scope of this study to prove that germ cell-independent transcriptional rewiring of the somatic supporting cells is also the case in other teleost species with TGF- β signaling-related sex-determining genes, we would like to imply it here. Our findings also suggest the small laboratory fish *Nothobranchius furzeri* as a suitable model to study a potentially general role of TGF- β signaling factors in the sexual differentiation of somatic supporting cells likely upstream of canonical mechanisms.

Of course, we are also aware that allelic diversification and gene duplication are the two mechanisms by which sex-determining genes usually evolve². However, for the first time, we show that a gene not only gained the sex-determining function but still maintains its original, not sex-related one likely due to its evolutionarily young age.

1. Wang R, *et al.* Dissecting Human Gonadal Cell Lineage Specification and Sex Determination Using A Single-cell RNA-seq Approach. *Genomics Proteomics Bioinformatics* **20**, 223-245 (2022).
2. Herpin A, Schartl M. Plasticity of gene-regulatory networks controlling sex determination: of masters, slaves, usual suspects, newcomers, and usurpators. *EMBO reports* **16**, 1260-1274 (2015).
3. Slanchev K, Stebler J, de la Cueva-Mendez G, Raz E. Development without germ cells: the role of the germ line in zebrafish sex differentiation. *Proc Natl Acad Sci U S A* **102**, 4074-4079 (2005).
4. Kurokawa H, *et al.* Germ cells are essential for sexual dimorphism in the medaka gonad. *Proc Natl Acad Sci U S A* **104**, 16958-16963 (2007).